# Spinful hinge states in the higher-order topological insulators WTe$_2$

Jekwan Lee[1,2], Jaehyeon Kwon[1,2], Eunho Lee[1,2], Jiwon Park[1,2], Soonyoung Cha[3,4], Kenji Watanabe [5], Takashi Taniguchi [5], Moon-Ho Jo [3,4] & Hyunyong Choi [1,2] ✉

Higher-order topological insulators are recently discovered quantum materials exhibiting distinct topological phases with the generalized bulk-boundary correspondence. $T_d$-WTe$_2$ is a promising candidate to reveal topological hinge excitation in an atomically thin regime. However, with initial theories and experiments focusing on localized one-dimensional conductance only, no experimental reports exist on how the spin orientations are distributed over the helical hinges—this is critical, yet one missing puzzle. Here, we employ the magneto-optic Kerr effect to visualize the spinful characteristics of the hinge states in a few-layer $T_d$-WTe$_2$. By examining the spin polarization of electrons injected from WTe$_2$ to graphene under external electric and magnetic fields, we conclude that WTe$_2$ hosts a spinful and helical topological hinge state protected by the time-reversal symmetry. Our experiment provides a fertile diagnosis to investigate the topologically protected gapless hinge states, and may call for new theoretical studies to extend the previous spinless model.

Recently, a new class of topological phase, called a higher-order topological insulator (HOTI), is proposed based on the generalized bulk-boundary correspondence, covering $d-2$ or lower-dimensional topological boundaries in $d$-dimensional systems[1–3]. For instance, time-reversal invariant three-dimensional (3D) HOTIs exhibit gapless hinge states, where the gapped surfaces are facing each other with a reversed sign of the mass. Such a phenomenon can be understood based on the fact that the gapped surface states host a doubly inverted electronic band and the strong spin-orbit coupling (SOC)[2,4–6]. With these physical grounds, the band structure and the corresponding topological features of HOTIs have been predicted by well-established methods such as a multi-orbital tight-binding model, first principle calculation, and Wilson loop calculation[1–6]. To date, there exist only a few condensed matter systems predicted to be HOTIs, such as bismuth[7,8], topological crystalline insulator SnTe[2,4,9], twisted bilayer graphene[10–12], and some artificial lattices[13,14].

Among such candidates, WTe$_2$ has recently attracted much interest in investigating the electronic correlations as well as exploring the topologically protected quantum phenomena[15,16]. With an orthorhombic 3D structure, it was first known as a type-II Weyl semimetal with electron and hole pockets around the Weyl points[5,17,18]; resolving the Weyl points, however, remains challenging because angle-resolved photoemission spectroscopy (ARPES) cannot provide sufficient momentum resolution to resolve the small separation of Weyl points of WTe$_2$[19,20]. In a monolayer limit, the thickness-dependent studies on the crystal symmetry and electronic band structure have revealed the quantum spin Hall insulating phase for 1T′-WTe$_2$ crystals[21,22]. After recent proposals on the higher-order topology, the large arc-like surface states of the bulk WTe$_2$, which were initially considered topologically trivial, started to be understood as gapped fourfold Dirac surface states[4]. Spatially resolved measurements using a Josephson junction were then used to identify the hinge states as a clue for the higher-order topology[7], and subsequent experiments have reported anisotropic confinement of 1D conducting hinge channels in few-layer $T_d$-WTe$_2$[23,24]. However, experimental evidence for the symmetry-protected topological nature of the observed 1D hinge state is still

[1]Department of Physics and Astronomy, Seoul National University, Seoul 08826, Korea. [2]Institute of Applied Physics, Seoul National University, Seoul 08826, Korea. [3]Center for Epitaxial van der Waals Quantum Solids, Institute for Basic Science, Pohang 37673, Korea. [4]Department of Materials Science and Engineering, Pohang University of Science and Technology, Pohang 37673, Korea. [5]Advanced Materials Laboratory, National Institute for Materials Science, 1-1 Namiki, Tsukuba 305-0044, Japan. ✉e-mail: hy.choi@snu.ac.kr

lacking. Moreover, even in a broader sense, a time-reversal invariant spinful feature of the helical HOTI in a natural solid-state system has not been investigated[25].

In this work, we experimentally show that atomically thin $T_d$-WTe$_2$ is indeed a time-reversal invariant HOTI hosting the helical spinful hinge states. To investigate the spin orientation of the hinge states, we have performed the spatially resolved polar magneto-optic Kerr-rotation measurements on WTe$_2$-graphene heterostructure devices. Our results agree with the previous spin-resolved observation of WTe$_2$, implying the possible gapless nature of the spin-polarized states[26,27]. In our measurements, the bulk- (or gapped surface-) and hinge-originated spin polarization can be distinguished by the Fermi level dependence of the Kerr rotation signals. Furthermore, we examine the time-reversal invariance of the spinful hinge states by opening the mass gap via external magnetic fields.

## Results

Multilayer $T_d$-WTe$_2$ has a noncentrosymmetric orthorhombic structure belonging to the SG 31 ($Pmn2_1$) space group with two perpendicular axes ($a$- and $b$-axis) and one mirror line along the $b$-axis (Fig. 1a). Together with the time-reversal symmetry, this spatial mirror symmetry satisfies necessary prerequisites to support the topologically non-trivial spin-polarized helical hinges[26,28]. In our experiments, multilayer WTe$_2$ is placed on monolayer graphene to detect the spinful 1D

hinge state by observing the spin polarization of electrons in graphene injected from WTe$_2$. The experiment schematic is illustrated in Fig. 1b. The bias voltage applied to the graphene channel forms a potential gradient to the bottom of the multilayer WTe$_2$, so the conducting electrons of WTe$_2$ are injected into the graphene. The spatial distribution of the spin-polarized electrons is recorded by the Kerr rotation microscopy with a submicrometer-scale resolution. Because the spin diffusion length is sufficiently long in single-layer graphene[29,30], we infer that the spin-polarized electrons in graphene contain the necessary spin information of WTe$_2$. Therefore, we interpret the differential Kerr rotation ($\Delta\theta_K$) in the scanning area, obtained by subtracting the Kerr rotation ($\theta_K$) at each spatial point with and without the bias voltage, as a manifestation of the electron spin polarization originated from WTe$_2$. Our device employs a tunable gate voltage ($V_G$) that enables us to distinguish the bulk and the hinge contribution by inspecting the Fermi-level-dependent $\Delta\theta_K$. An optical microscopy image of a complete device is shown in Fig. 1c with the crystal $a$- and $b$-axis, where it is designed to perform the electrical and optical measurements along both axes. The crystal axes were verified by measuring the polarization-dependent absorption, as shown in Fig. 1d.

We start by presenting the $V_G$-dependent Kerr-rotation signals to investigate the spinful characteristics of the anisotropic WTe$_2$ hinge states. Figure 2a shows the transfer curve between contact 1 and 3, i.e., parallel to the $a$-axis referring to Fig. 1c. The observed two

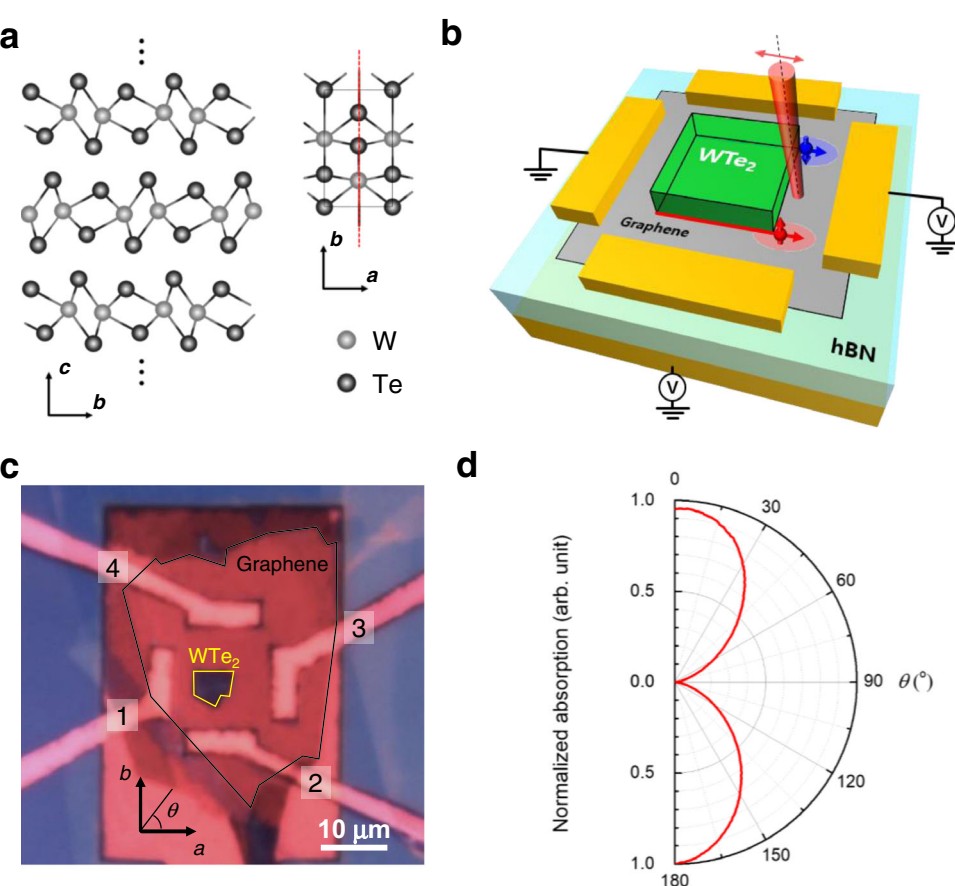

**Fig. 1 | Crystal structure of multilayer Td-WTe2 and experimental design. a** The $T_d$ structure of multilayer WTe$_2$ is non-centrosymmetric with a mirror plane $M_a$ (red dashed line). **b** Schematic experimental design for detecting the spin-polarized electronic states in WTe$_2$. The electrical bias voltage makes electrons flow through WTe$_2$, while the spin polarization of the electrons is optically recorded as the Kerr rotation induced in the linearly polarized pump (980 nm, 1,415 W/cm²). The pump laser with a spot size of 1.5 μm sweeps through a 6 μm × 10 μm region at graphene near the edge of WTe$_2$ by scanning mirrors to obtain the spatially resolved Kerr

rotation data. **c** An optical microscopy image of the device is shown. The multilayer WTe$_2$ (yellow) and monolayer graphene (black) are highlighted. Electrodes are labeled as contact numbers 1, 2, 3, and 4. **d** A normalized polar plot of the polarization-dependent absorption of the multilayer WTe$_2$ is shown. The absorption was measured at the center of the WTe$_2$ flake in the device while varying the polarization of 980 nm laser light. The anisotropy of the absorption indicates that the crystal axes are placed as shown in **c** (black arrows).

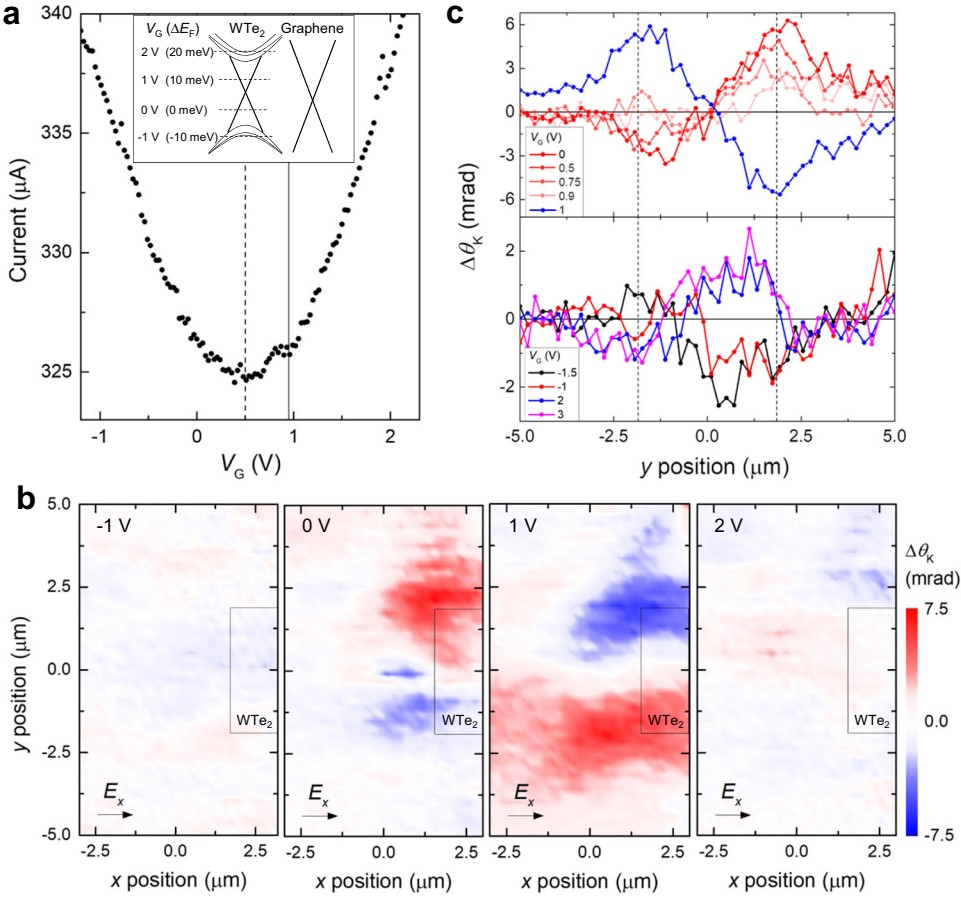

**Fig. 2 | Gate voltage dependence of the spatially resolved differential Kerr rotation. a** The $V_G$-dependent drain current parallel to the $a$-axis of WTe$_2$ is shown. The measurements were performed at 1.6 K. The longitudinal bias voltage was 0.5 V between contact 1 and 3 (parallel to the $a$-axis; see Fig. 1c for the contact number). Two charge neutral points were observed; one at $V_G$ = 0.5 V is for graphene (black dashed line) and another at $V_G$ = 0.95 V is for WTe$_2$ (black line). The illustration in the inset shows the schematic band alignment of graphene and WTe$_2$. Representative $V_G$ and the corresponding Fermi level change $\triangle E_F$ are marked as the black dashed lines. **b** Spatially resolved contour plots of the $V_G$-dependent $\Delta\theta_K$. The colors represent the spatially resolved $\Delta\theta_K$ at $V_G$ = -1, 0, 1, 2 V. The bias voltage of 0.5 V is applied between contact 1 and 3 to form a longitudinal electric field in +$x$ direction (parallel to the $a$-axis; see Fig. 1c for the contact number). The black rectangle in each plot denotes the left end part of the WTe$_2$ flake. **c**, Line-cut plots of $\Delta\theta_K$ at $x$ = 0.75 μm are shown. The $V_G$-dependent $\Delta\theta_K$ in the top panel (0 ≤ $V_G$ ≤ 1 V) shows the localized $\Delta\theta_K$ near $y$ = ±1.85 μm; these $y$ positions correspond to the WTe$_2$ hinge location (black dashed lines). The $V_G$-dependent $\Delta\theta_K$ for $V_G$ = −1.5, −1, 2, 3 V are displayed in the bottom panel.

conductance deeps at $V_G$ = 0.5 and 0.95 V correspond to the charge neutrality point of the graphene and the multilayer WTe$_2$, respectively, as illustrated in the inset of Fig. 2a. Two-dimensional (2D) contour plots in Fig. 2b display the spatially resolved $\Delta\theta_K$ near the WTe$_2$ edge with varying $V_G$ ($V_G$ = −1, 0, 1, 2 V) in the absence of the external magnetic field. At $V_G$ = 0 and 1 V, a substantial amount of the spin-polarized electrons is concentrated near $y$ = ±1.85 μm, while $\Delta\theta_K$ is evenly distributed throughout $|y|$ ≤ 1.85 μm at $V_G$ = -1 and 2 V. Considering the $V_G$-tuned Fermi level and the spatial arrangement of $\Delta\theta_K$, the observed $\Delta\theta_K$ distributions at $V_G$ = 0 and 1 V match the spin-polarized in-gap states localized in the hinge, while those of $V_G$ = -1 and 2 V represent the electrons from the spin-split bulk bands. The opposite sign of $\Delta\theta_K$ seen near the two parallel hinges indicates the spinful and helical nature of the localized electron states. To elucidate the bulk- and hinge-originated $\Delta\theta_K$ in detail, we show in Fig. 2c the line-cut plots of $y$-dependent $\Delta\theta_K$ measured at different $V_G$. In the bulk-insulating range of $V_G$ (Fig. 2c, top panel), $|\Delta\theta_K|$ localized at $y$ = 1.85 μm decreases monotonically with increasing $V_G$, and $\Delta\theta_K$ changes the sign abruptly when $V_G$ reaches 1 V. This change is consistent with Fig. 2a, where $V_G$ = 1 V is above the charge neutral point. On the other hand, when WTe$_2$ is degenerately doped, i.e., $V_G$ ≥ 2 V or $V_G$ ≤ -1 V (Fig. 2c, bottom panel), $\Delta\theta_K$ evenly spreads across $|y|$ ≤ 1.85 μm, and no sign change of

$\Delta\theta_K$ across the $y$ position was observed. Note that the sign of $\Delta\theta_K$ implies the orientation of spin-polarized electrons, and $|\Delta\theta_K|$ denotes the concentration of the conducting electrons (or the density of state at the Fermi level, equivalently) with the corresponding spin. These results strongly suggest that although the spin configuration of helical hinge states of the bottom surface of multilayer WTe$_2$ resembles that of the spin-momentum-locked helical edge states of the 2D quantum spin Hall insulator. Note that the multilayer WTe$_2$ is not simply a stack of weak 2D topological insulator layers as proven previously[23].

As for the HOTI characteristics, we note that the band topology of the multilayer WTe$_2$ should be protected by the time-reversal symmetry. One method to examine such topological protection, which is associated with the gapless band with degenerated Dirac points, is to perform the magnetic-field dependent $\theta_K$ measurements. Figure 3a–c show the line-cut plots of the $V_G$-dependent $\theta_K$ under external magnetic field $B_z$ of 0.5, 1, and 2 T, applied perpendicular to the device $xy$ plane. In Fig. 3a, where $B_z$ is 0.5 T, we see that the $\Delta\theta_K$ near the hinges vanishes as $V_G$ approaches the charge neutrality. With increasing $B_z$ of 1 T (Fig. 3b), the localized $\Delta\theta_K$ survives only when $V_G$ is pushed further below (0 V) and above (1.5 V) the charge neutrality point. When $B_z$ is sufficiently large, Fig. 3c shows the vanishing $\Delta\theta_K$ signals, which imply that no spin-polarized electrons are present; this can be readily

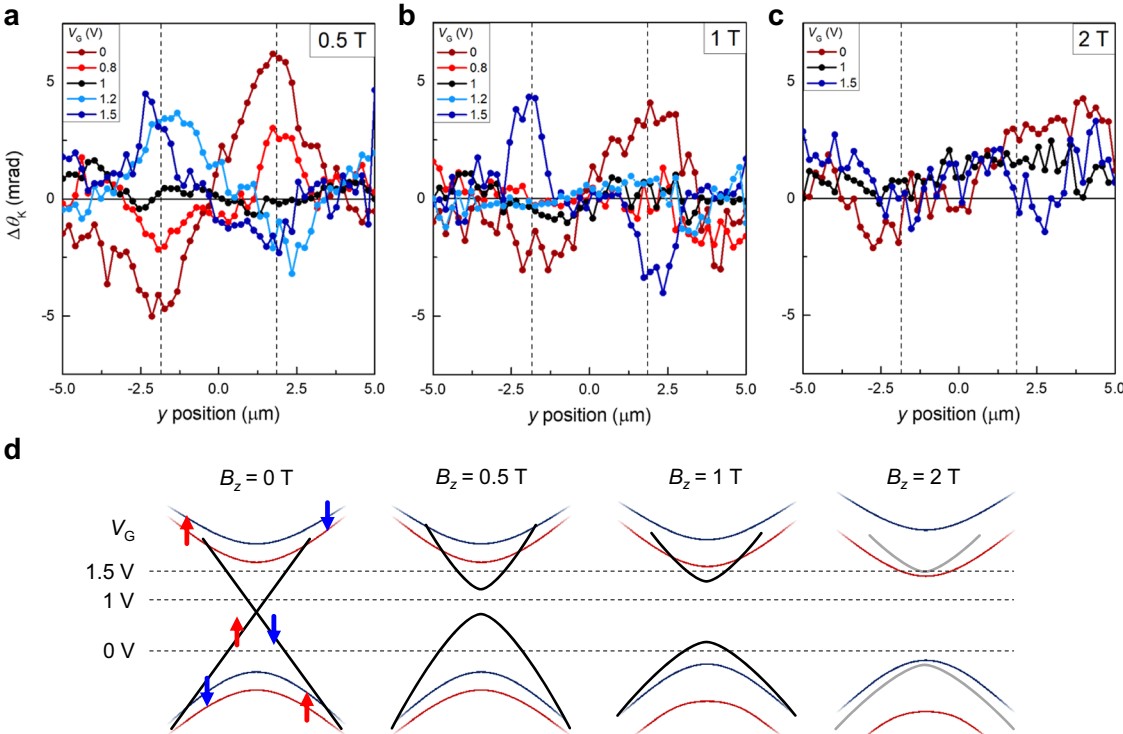

**Fig. 3 | Gap opening of the multilayer WTe2 due to broken time-reversal symmetry. a–c** The line-cut plots show $\Delta\theta_K$ at $x = 0.75\,\mu m$ with varying $V_G$ under $B_z = 0.5\,T$ (**a**), $1\,T$ (**b**), and $2\,T$ (**c**). $\Delta\theta_K$ is featureless only at $V_G = 1\,V$ when $B_z = 0.5\,T$, while it shows no variation when the applied $V_G$ is $0.8\,V \leq V_G \leq 1.2\,V$ under $B_z = 1\,T$. Note that no localized $\Delta\theta_K$ behavior is seen at any $V_G$ when $B_z = 2\,T$. Dashed lines in **a–c** at $y = \pm 1.85\,\mu m$ indicate the $y$ position of the WTe2 hinges in the real space. **d**, Schematic band structures representing the effect of $B_z$ on the spin-polarized hinge states (black lines) and spin-split bulk bands (colored lines). Because $B_z$

breaks the time-reversal symmetry, Dirac fermions at the topological hinge states gain an effective mass. This opens a finite energy gap, which is proportional to the magnitude of $B_z$. The gap opening appears as a flat $\Delta\theta_K$ along $y$ since the Fermi level falls within the gap. The dashed lines in the diagram indicate the Fermi levels when $V_G$ is 0, 1, and 1.5 V. For the case when $B_z = 2\,T$ (**d**), the schematic represents one possibility that the hinge states are merged into the bulk band due to the induced gap in the hinge states.

understood as the mass gap opening due to the broken time-reversal symmetry[31,32]. The schematic diagrams in Fig. 3d show how $B_z$ is expected to affect the gapless dispersion of the helical hinge states. Without $B_z$, the hinge states remain gapless because the degeneracy of the Dirac point is protected by time-reversal symmetry. In the case of relatively weak $B_z$ (= 0.5, 1 T), the hinge opens a bandgap while preserving its spin texture (Fig. 3a, b). On the other hand, when a relatively strong $B_z$ of 2 T is applied (Fig. 3c), the trace of the hinge disappears. Such disappearance of $\Delta\theta_K$ characteristics when $B_z = 2\,T$ may originate from either the hinge states being merged into the bulk while maintaining the HOTI phase (Fig. 3d)[33] or WTe2 exhibits no HOTI phases with increasing external magnetic fields. A further theoretical investigation is necessary to elucidate the correlation between the spin texture and the band configuration under strong $B_z$.

There might exist alternative scenarios on the role of $B_z$ other than the mass gap opening. First, one plausible explanation would be the formation of quantum Hall states accompanied by the chiral boundary. However, $B_z$ used in our experiment is not strong enough to generate such an effect[18,34], and the observed $V_G$-dependent counter-propagating hinge modes are not consistent with the chiral state characteristics. Second, the effect of $B_z$ on the graphene channel may cause a similar $V_G$ dependence of $\Delta\theta_K$, such as opening a gap or causing transverse spin (or valley) flow in graphene. However, existing studies show that $B_z$ of 10 T is the lower boundary to observe such effects, which is far larger compared to our $B_z$[35-37]. Lastly, the broken time-reversal symmetry can be associated with the spatial split of the hinge modes rather than the bandgap opening[32]. In our experiment regime, the applied $B_z$ makes the hinge a boundary between one parallel to $B_z$ and another perpendicular to $B_z$. Thus, considering the non-zero mass

and Zeeman contribution to the position away from the hinge, such spatially shifted hinge modes cannot occur between the two surface states.

The transverse spin accumulation originated from the WTe2 bulk, i.e., spin Hall effect (SHE), might be an alternative to explain the observed $\Delta\theta_K$. To further substantiate that observed $\Delta\theta_K$ features arise from the spinful hinge state exclusively, we investigated the spatially resolved $\Delta\theta_K$ using a device with a modified structure (device #4). Figure 4a shows the corresponding optical microscopy image. The graphene layer below the multilayer WTe2 has a 1.5 μm wide gap along the *a*-axis of the multilayer WTe2 (see Supplementary Note 1-3 for details). If $\Delta\theta_K$ originates from the hinge states, the localized $\Delta\theta_K$ should arise only near the WTe2 hinge (Fig. 4b). On the other hand, if the observed $\Delta\theta_K$ originates from the bulk spin transport in WTe2, the spin-polarized electrons injected into graphene are expected to be spread out to the left as well as to the right of the graphene area in a transverse direction to the applied electric field (Fig. 4c). Therefore, the presence of a gap in graphene would collect the accumulated spin-polarized electrons at the edge of graphene on both sides of the gap (Fig. 4c). To check the above idea, we have investigated the magneto-optic Kerr effect on device #4. The results are shown in Fig. 4d, e. Here we measured the spatially resolved $\Delta\theta_K$ at different $V_G$ with and without an external magnetic field ($B_z = 1\,T$) (see Supplementary Note 4 and Figs. S15, 16 for spatially resolved $\Delta\theta_K$ with $B_z$). We first note that no accumulation of the spin-polarized electrons was seen on either side of the graphene gap, regardless of $V_G$. Secondly, with varying $V_G$ (see Supplementary Note 1-2 for the relationship between $V_G$ and $\Delta E_F$), Fig. 4d, e shows that $\Delta\theta_K$ appears only in line with the WTe2 hinges. We also observed a clear sign flip of $\Delta\theta_K$ when $E_F$ is swept across the

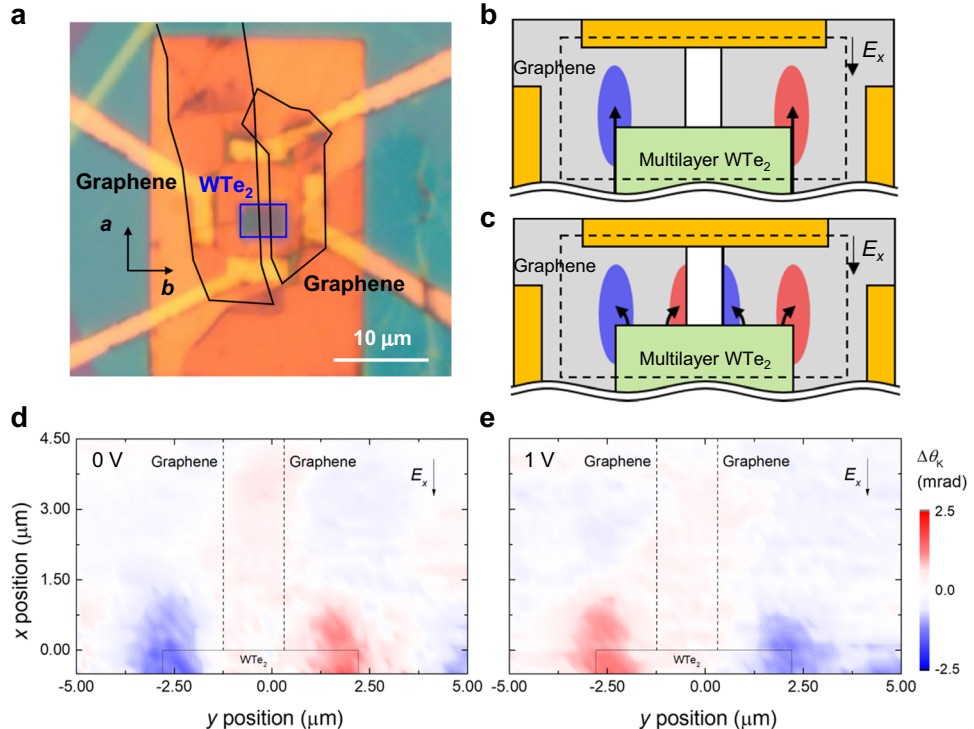

**Fig. 4 | Spatially resolved differential Kerr rotation on a device with a spatial gap in graphene. a** An optical microscopy image is shown. Two monolayer graphene flakes are separated by a 1.5 μm gap. This device scheme is almost identical to the other devices, except the presence of a gap in graphene. The graphene layer for the electron transport measurement is located below $WTe_2$. **b, c** Schematic diagrams of the expected $\Delta\theta_K$ when the spin-polarized electrons are injected in graphene from the hinges (**b**) and when they originate from the bulk (**c**). Dashed rectangles indicate the window of spatially resolved measurement, and black arrows indicate electron transport. **d, e** Contour plots of $\Delta\theta_K$ observed in device #4 when $V_G = 0$ (**d**), 1 V (**e**) are shown. The $V_G$-dependent transport measurements are shown in Fig. S4. The distribution of $\Delta\theta_K$ is as expected in **b**, meaning there was no spin accumulation at the edge of graphene, and the spin-polarized electrons are originated from the hinges of multilayer $WTe_2$. Dashed lines mark the edge of each graphene layer, and the black rectangles indicate the location of the multilayer $WTe_2$ flake.

Dirac point of the hinge states. Third, $\Delta\theta_K$ under the magnetic field (Figs. S15, 16) shows the gap opening of the hinge states. Under the external magnetic field $B_z$ of 1 T, the localized $\Delta\theta_K$ at the $y$-position of the hinges disappears when the Fermi level is close to the Dirac point (i.e., $V_G$ near 0.88 V in the case of device #4), demonstrating the lifted degeneracy of hinge eigenstates due to the broken time-reversal symmetry. To summarize, the $V_G$- and $B_z$-dependent $\Delta\theta_K$ distribution in device #4 is essentially identical to the devices without the graphene gap (see Fig. 2 and Figs. S6–11). These data provide additional evidence that SHE is not likely the origin of our observation.

In conclusion, we experimentally have shown that the multilayer $T_d$-$WTe_2$ is a time-reversal invariant helical HOTI possessing the spinful hinge states. The spin polarization of electrons originating from the 1D hinge state of the multilayer $WTe_2$ was investigated by the spatially resolved magneto-optical Kerr rotation measurement in the $WTe_2$-graphene heterostructure device. The $V_G$- and $B_z$-dependent data provide strong evidence that the helical spin-polarized states are within the bulk bandgap while they are localized at the geometric hinge of the multilayer $WTe_2$, whose energy degeneracy is protected by the time-reversal symmetry. Because the topologically protected spinful mode is highly confined in the 1D channel, the hinge state of the HOTI may open up a new arena to study the strong correlation and topology in other higher-order topological materials.

## Methods
The detailed information about the device fabrication, experimental setup and full dataset with further discussion are available in Supplementary Information.

## Data availability
The data that support the findings of this study are available from the corresponding author on reasonable request.

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

## Acknowledgements

J.L., J.K., J.P., and H.C. were supported by the National Research Foundation of Korea (NRF) through the government of Korea (Grant No. NRF-2021R1A2C3005905, NRF-2020M3F3A2A03082472), Creative materials Discovery program (Grant No. 2017M3D1A1040828), Scalable Quantum Computer Technology Platform Center (Grant No. 2019R1A5A1027055), the Ministry of Education through the core center program (2021R1A6C101B418), and the Institute for Basic Science (IBS), Korea, under Project Code IBS-R014-G1-2018-A1. SC and M.-H.J were supported by the IBS, Korea, under Project Code IBS-R014-A1. Part of this study has been performed using facilities at IBS Center for Correlated Electron Systems, Seoul National University.

## Author contributions

J.L., J.K., E.L. fabricated samples. J.L., J.P., S.C. performed the device characteristics examination. K.W. and T.T. provided high-quality hBN crystal. J.L., M.-H.J., and H.C. performed data analysis and discussed the results. H.C. supervised the project. J.L. and H.C. wrote the manuscript with input from all co-authors.

## Competing interests
