## [Peer Review File · Nature Communications]

Reviewers' Comments:

Reviewer #1:

Remarks to the Author:

The manuscript reported the observation of spinful hinge states in the multilayer Td-WTe₂ by employing the magneto-optic Kerr effect and examining the spin polarization of electrons injected from WTe₂ to graphene under external magnetic fields.

However, by analyzing the data in this manuscript, it does not seem to be able to give clear evidence for the existence of the hinge states. The reasons are as follows:

1. From the data and energy bands in Figs. 2 and 3 and the explanations, the contributions from these claimed hinge states of higher-order topological insulators (HOTIs) are more likely to come from the helical edge states of conventional TIs. One knows that the single-layer WTe₂ is experimentally confirmed conventional TI. Assuming that Td-WTe₂ is HOTI, its hinge states should exist in the gap of both bulk and gaped surface states. This gap cannot be of the order of eV, but should be in the meV magnitude.
2. An important observation is that the differential Kerr rotation is only observed along the a-axis but not the b-axis. Is it possible that this comes from the strong in-plane anisotropy of WTe₂, as shown in Fig. 1 (d), caused by different symmetries, or related to the Berry curvature dipole?
3. Figure 2 (b) shows the spatially-resolved Kerr rotation. At a gate voltage of 0eV, especially 1eV, the signal is distributed over a large area (red area between -2.5 μm —2.5 μm), which is not consistent with the distribution of possible hinge states along the a-axis (near the top and bottom edges of the black rectangles). In addition, why did the authors choose a line cut with $x=0.75 \mu\text{m}$? This is some distance away from the area of possible hinge states.

In addition to the above general comments, there are some suggestions

1. The thickness of the multilayer Td-WTe₂ used in the manuscript is 4nm (8 layer). In order to distinguish different possible topological phases in Td-WTe₂, such as Weyl semimetal, HOTI and 2D TI from 3D to 2D. The data of samples of different thickness are needed.
2. In the caption of Fig. 2, "VG = -1.5, -1.2, 3 V are displayed" should be "VG = -1.5, -1, 2, 3 V are displayed".

In summary, I cannot be persuaded to recommend the manuscript in Nat. Commun..

Reviewer #2:

Remarks to the Author:

The experimental study reported in this manuscript is a well-motivated and timely one, aimed at finding experimental evidence for the helical Dirac nature of higher order topological insulator hinge states that are predicted to exist in Td-WTe₂. This is a challenging and ambitious goal since the usual approach of spin- and angle- resolved photoemission does not have the spatial and energy resolution to probe the spin polarization of these states. This manuscript reports a clever idea for observing the putative spinful hinge states via spatially-resolved polar magneto-optic Kerr effect (MOKE) measurements on very thin WTe₂-graphene heterostructure devices and claims that the observations provide evidence for spinful hinge states. The experiment is carried out in the following geometry: a flake of WTe₂ is placed in the middle of a larger graphene device that carries a current along either the a axis or the b axis. The entire device can be back gated, allowing the Fermi energy to be varied in both the graphene and WTe₂ layers. In the region of the WTe₂ flake, this current flows in parallel through the graphene and WTe₂ channels. If spinful hinge states exist, this should result in a spin accumulation at the edges of the WTe₂ and create a spin polarization that should diffuse into the vicinal graphene. Spatially-resolved MOKE measurements then probe the spin polarization in the device using the Kerr rotation. This shows that when the current flows along the a axis, a spin polarized signal is observed via MOKE in the graphene near

the edges of the WTe2 flake. A key feature here is that the MOKE signal is strongly dependent on the back gate voltage (V_g) and changes sign as the Fermi energy is swept through the charge neutral point of WTe2. The signals are quite small but the signal-to-noise is good enough to produce reasonably convincing signatures of spin polarization. The authors also study the MOKE signal as a function of magnetic field and argue that its disappearance at higher fields is consistent with the requirement of time reversal symmetry for the presence of spinful hinge states. The authors do a reasonably systematic job of ruling out some obvious explanations (e.g. Landau levels, quantum Hall effect). I note that in principle, just a regular spin Hall effect in WTe2 could also produce such signals by sourcing spin current into the graphene. I believe that this would also change sign as the Fermi energy passes through the charge neutral point. The authors carry out a measurement with the current flowing along the b axis to rule this out, showing that a localized spin polarization is not observed near the edges. However, as the authors correctly note, the spin-orbit coupling is also much weaker for this case. So, I am not convinced that one can rule out the conventional spin Hall effect and immediately jump to the conclusion that the observed MOKE signal is arising from spinful hinge states in WTe2. (I do not understand the data shown as "hinge" and "bulk" in Fig. 4b -- the paper does not explain whether this is taken for current flow along the a axis or b axis -- I guess it is the former but it's confusing since Fig. 4a focuses on current flowing along the b axis.)

I find all these observations to be very intriguing, interesting, and puzzling. It's an elegant experiment, ambitious in its goal, and it would be worthwhile publishing this data in a high profile journal such as Nature Communications. However, the manuscript in its current form has some major deficiencies.

First, I always find it troublesome to see an experimental manuscript that only shows data from a single device. It's crucial to show that the observations can be reproduced and give the reader some idea of how readily this experiment can be repeated. Was this one device out of many in which these observations could be seen? Or do the authors have data on additional devices? Why not show all the data in the supplementary section and make a comment on the statistical device-to-device variations?

Second, it is important to show some control experiment. For example, is there a way to make a device wherein the WTe2 layer is not predicted to host spinful hinge states? Does the experiment show the complete absence of a signal in that case?

Finally, while the paper is reasonably well-organized and easily followed by a specialist who is interested enough to ignore the superficial language issues, the paper could be more carefully edited to improve the readability for non-specialists. Below, I give some examples of sentences that are problematic.

Until all the deficiencies that I have outlined above are taken care of, especially measurements on additional devices and a control, even with the very interesting data obtained on the devices shown in the current manuscript, I cannot recommend that this paper be published in Nature Communications.

A few examples of awkward or grammatically incorrect language:

Abstract:

"Higher-order topological insulator is a recently discovered topological phase of matters with generalized bulk-boundary correspondence."

"Our finding may invoke stimuli to extend the previous spinless models and further provide a fertile diagnosis to the higher-order topology and Weyl physics in multidimensional solids."

Page 3: "The elemental origins of the gapped surface state are the..."

Page 3: "Among such candidates, WTe2 has drawn crucial interests in the perspective of band 1 topology."

Page 6: "One seeming method to examine such topological protection..."

Page 7: "Without B_z , the hinge states host the gapless distribution with Dirac points protected by the time-reversal symmetry."

Page 7: "With increasing $B_z \geq 0.5$ T, the oscillatory $\delta\text{-}\theta_K$ subdue in the broader range of VG."

Page 7: "...or WTe2 exhibits no the HOTI phase."

Page 8: "...the new massless position cannot occur between two surfaces states, which ascertains the mass gap open in our experiments."

Reviewer #3:

Remarks to the Author:

- What are the noteworthy results?

The results of note are the claim to direct observation of hinge states in WTe2.

- Will the work be of significance to the field and related fields? How does it compare to the established literature? If the work is not original, please provide relevant references. According to the authors, there already exists evidence for hinge states from an experiment where WTe2 was coupled to a superconductor. The claim here is that this experiment is a "direct observation of hinge states" and resolves the "spinful" nature of the transport. If correct, this would be a significant contribution. However, as I discuss below, I am not at all clear on the evidence supporting these conclusions.

- Does the work support the conclusions and claims, or is additional evidence needed?

I have some questions about the fundamentals of the experiment that are not explained in the manuscript. The most basic is how the authors interpret the asymmetric spin accumulation that they observe. I understand the hinge states to be 1D channels at the surface. The spin polarization of the hinge states is not clear to me. Does each side of the surface support two counterpropagating spin up and down modes, as in the Quantum Spin Hall Effect? It would be very useful to clarify this point.

If this is indeed the case, it is not clear to me how to account for spin polarization on the basis of hinge states alone. The asymmetric spin accumulation presumably arises from a spin current transverse to the current provided by the voltage bias. Isn't it necessary that this current arise from the bulk in as much as the hinge states support only purely longitudinal currents?

Based on this, I would tend to argue that the bulk spin Hall Effect (SHE) must be involved in generating the spin accumulation that is observed. The authors appear to claim is that this effect is ruled out by two observations. The first is that the spin accumulation is not seen when the Fermi level is set to the bulk states of WTe2. However, the identification of the position of the Fermi level is very unclear from the data as the feature at 0.95 eV is barely above the noise level. Is this feature reproducible? Moreover, it is unclear to me why the SHE should not be present in bulk states of WTe2 in as much as it is allowed by symmetry and is proportional to spin-orbit coupling, which is strong in this material. Finally, is it just a coincidence that the overall amplitude of the spin accumulation in the two putative regimes of conduction only differ by a factor of about 2? The second piece of evidence is the difference in spin accumulation for the two directions of bias current. However, the authors undercut this evidence themselves by pointing out that that "WTe2 possesses a relatively small SOC along the b-axis than the a-axis."

- Are there any flaws in the data analysis, interpretation and conclusions? Do these prohibit publication or require revision?

In answer to this question, let me repeat my basic lack of understanding of what is being claimed here. I don't see how one explains spin accumulation at the edges of a sample without transverse spin current. Perhaps I am missing something – if this could be explained the paper would be on firmer ground. However, in view of lack of spatial resolution, the claim to direct observation should be softened.

- Is the methodology sound? Does the work meet the expected standards in your field?

I worries me a bit that many of the observations such as the neutrality points for the graphene

and WTe₂, and the polar Kerr signal itself, are so close to the noise limit. It would be useful to see some evidence for the reproducibility of these effects. Another point regarding methodology is that I don't understand why the polar Kerr effect is observed in the graphene rather than in the WTe₂ itself.

- Is there enough detail provided in the methods for the work to be reproduced?

I believe there is enough detail in this regard.

Point-by-point responses to the issues raised by the reviewers

General remarks and comments of Reviewer 1: The manuscript reported the observation of spinful hinge states in the multilayer Td-WTe₂ by employing the magneto-optic Kerr effect and examining the spin polarization of electrons injected from WTe₂ to graphene under external magnetic fields. However, by analyzing the data in this manuscript, it does not seem to be able to give clear evidence for the existence of the hinge states. The reasons are as follows:

Response 1 general remarks: We appreciate the time that Reviewer 1 took to consider our manuscript. Reviewer 1 raised concerns about other possible scenarios in explaining our experiments. We have carefully considered these points mentioned by Reviewer 1 and thoroughly revised our manuscript by providing additional experimental data to support our main idea. We also made corrections on some expressions, which may cause misunderstandings.

As one of our major efforts in preparing the responses, we fabricated a few more devices (device #2 - device #4) and checked the reproducibility of experiments with all the devices (Comment 1-4). An additional device (device #4) was fabricated with a slightly different geometry. This was used to demonstrate that the spin Hall effect (SHE) is not likely the origin of our observed $\Delta\theta_k$ distribution (Comment 1-2).

Please note that we have moved Extended data and Methods in the original manuscript to Supplementary Information in the revised manuscript; these include the data from additional experiments conducted with newly fabricated devices.

Below we present our point-by-point responses to Reviewer 1's comments.

Comments 1-1: 1. From the data and energy bands in Figs. 2 and 3 and the explanations, the contributions from these claimed hinge states of higher-order topological insulators (HOTIs) are more likely to come from the helical edge states of conventional TIs. One knows that the single-layer WTe₂ is experimentally confirmed conventional TI. Assuming that Td-WTe₂ is HOTI, its hinge states should exist in the gap of both bulk and gaped surface states. This gap cannot be of the order of eV, but should be in the meV magnitude.

Response 1-1: We apologize for the oversimplified discussion on Figs. 2 and 3. As Reviewer 1 has commented, we are also aware that the energy gap of bulk and surface of the multilayer T_d-WTe₂ is the order of the meV scale [Phys. Rev. Lett. **123**, 186401 (2019); Nat. Mater. **19**, 974-979 (2020)]. The misunderstanding possibly arises from the fact that we have not clearly explained the gate-dependent Fermi-level changes ΔE_F in the electronic transport data, especially the inset of Fig. 2; we have made corrections to the inset as shown below in the last part of Response 1-1.

The device we have used is a simple bottom-gated one, where we have used an hBN layer of thickness $d = 25$ nm as a gate insulator and a metal (Ti 5 nm, Au 25 nm) electrode as a bottom gate. We have estimated the Fermi-level change ΔE_F with respect to the applied bottom gate voltage V_G , where ΔE_F is a differential change of E_F compared to $V_G = 0$ V, using the following simple electrostatic calculation [Nat. Phys. **14**, 900-906 (2018)].

$$Q_{2D} = C_i V_i.$$

Here, $Q_{2D} = eA \int_0^{E_F} \frac{2|E|}{\pi \hbar^2 v_F^2} dE$ is the total charges accumulated in graphene with area A due to the gate voltage, $C_i = \varepsilon \frac{A}{d}$ is the capacitance of the graphene-hBN-electrode capacitor, and V_i is the gate voltage applied across the gate insulator hBN. The Fermi-level change ΔE_F was estimated using the following parameters: the Fermi velocity of graphene near Dirac point $v_F \sim 1 \times 10^6$ m/s [Phys. Rev. Lett. **101**, 096802 (2008)], the permittivity of hBN $\varepsilon = 3.75 \varepsilon_0$ [NPJ 2D Mater. Appl. **2** 6 (2018)], and the thickness of hBN $d = 25$ nm. From these values, the changes in Fermi level ΔE_F induced by the gate voltage V_G were estimated. For example, V_G of -1 and 2 V corresponds to -10 and 20 meV, respectively.

Based on the above calculation, when the sweep range of V_G is small enough, i.e., 0 V and 1V, it covers the hinge states within the expected bulk and surface gap (~ 20 meV) [Nat. Mater. **19**, 974-979 (2020); Phys. Rev. Lett. **123**, 186401 (2019)]. When V_G is large, i.e., -1 V

and 2 V, it introduces degenerately doped holes ($V_G = -1$ V) and electrons ($V_G = 2$ V), which are sufficiently large to screen the characteristic signatures of the hinge states.

We admit that the inset of Fig. 2 was very confusing. Referring to the black dashed lines in the inset, we should have correctly indicated the V_G -dependent ΔE_F . Not only Reviewer 1's comment here, but also another one (Comments 1-3, "gap cannot be of the order of eV") is because of such a confusing presentation. To avoid such confusion, we have modified the inset of Fig. 2a as follows.

Figure R1. Revised inset of Fig. 2a in the main text. The estimated value of ΔE_F corresponding to each V_G is added to clarify the relationship between them.

We have added the correlation between the range of the bottom gate voltage and the Fermi level in the revised main text and Supplementary Information. Figure 2 and corresponding explanations in the revised main text are edited to include such changes. In the Supplementary Information, Supplementary Note 1-2 provides how we estimated the relationship between V_G and ΔE_F .

Comments 1-2: 2. An important observation is that the differential Kerr rotation is only observed along the a -axis but not the b -axis. Is it possible that this comes from the strong in-plane anisotropy of WTe₂, as shown in Fig. 1 (d), caused by different symmetries, or related to the Berry curvature dipole?

Response 1-2: We thank Reviewer 1 for this critical comment. Indeed, it is important to consider other possible explanations for the observed Kerr rotation signals. We fully agree that it is crucial to rule out other effects experimentally. To accommodate this comment, we have fabricated a new device having a different geometric structure (device #4) and have performed the same Kerr rotation measurements. Please note the comments of Reviewer 2 (Comment 2-1) and Reviewer 3 (Comment 3-1), where both reviewers have also raised similar concerns about the alternative origin of $\Delta\theta_k$, especially SHE. Below we present our responses to such a similar question by providing new data obtained from a newly fabricated device.

First of all, our experiments investigated the hinge characteristics both along the a -axis and the b -axis. As shown in revised Supplementary Note 3-2 and Fig. S8, we found no evidence for the localized hinge states along the b -axis. This result is consistent with the prior studies [Nat. Mater. **19**, 974-979 (2020); Nano Lett. **20**, 4228-4233 (2020)], in which the Josephson transport experiments show that the Fausnhofer interference patterns of supercurrent distribution appear exclusively on the hinge state along the a -axis.

In our study, it is possible that the localized differential Kerr rotation $\Delta\theta_k$ arises from SHE due to the strong spin-orbit coupling in WTe₂. In this case, the spin-polarized electrons would be spatially distributed more along the transverse direction to the applied electric field (along the y -direction, Fig. R2a) rather than along the parallel direction (along the x -direction, Fig. R2b). However, data shown in Fig. 2b in the revised manuscript clearly show that $\Delta\theta_k$ is an elliptical shape whose major axis of the current distribution is inclined parallel to the applied electric field direction. Please note that separate discussions were made in our Response 1-3, where we provide why $\Delta\theta_k$ is spread over the WTe₂ boundary rather than being localized near the hinges.

Figure R2. Schematic diagrams representing the expected $\Delta\theta_k$ distribution when the spin-polarized electrons detected in graphene are injected from the bulk of WTe_2 , i.e., due to SHE (a), and when they are injected from the hinges of WTe_2 (b). The dashed rectangles are the spatial window that we performed the Kerr rotation measurements. Red, blue and black arrows represent the spin-polarized electronic transport of two opposite spins, respectively.

To provide additional evidence on the above discussion as well as to address Reviewer's concern, we have fabricated a new device (device #4) having almost the same structure but with a $1.5 \mu\text{m}$ gap in the middle of graphene. The optical microscopy image of the device is shown in Fig. R3a. If $\Delta\theta_k$ originates from the hinge states, the localized $\Delta\theta_k$ should arise only at the y-position of the WTe_2 hinge (Fig. R3b). On the other hand, if the observed $\Delta\theta_k$ originates from the bulk spin transport in WTe_2 (i.e., due to SHE), the spin-polarized electrons injected into graphene are expected to be spread out to the left as well as to the right of the graphene area in a transverse direction to the applied electric field (Fig. R3c). Therefore, the presence of a gap in graphene would collect the accumulated spin-polarized electrons at the edge of graphene on both sides of the gap (Fig. R3c).

Figure R3. An experiment proposed to isolate the hinge characteristics (device #4). **a** An optical microscopy image is shown. Two monolayer graphene flakes are separated by a $1.5 \mu\text{m}$ gap. This device scheme is almost identical to the other devices, except for the presence of a gap in graphene. The graphene layer for the electron transport measurement is located below WTe_2 . **b, c** Schematic representation for the expected $\Delta\theta_{\mathbf{k}}$ distribution when the spin-polarized electrons in graphene originate from the WTe_2 hinge states (**b**) and when they originate from the WTe_2 bulk (**c**). Dashed rectangles are the spatial windows that we performed the Kerr-rotation measurements. The black arrows are to represent electron transport.

To check the above idea, we have investigated the magneto-optic Kerr effect on device #4. The results are shown in Fig. R4. Here we performed the measurements of the spatially resolved $\Delta\theta_{\mathbf{k}}$ at different V_G with (Fig. R4a) and without (Fig. R4b) an external magnetic field ($B_z = 1$ T). We first note that no accumulation of the spin-polarized electrons was seen on either side of the graphene gap, regardless of V_G . Secondly, with varying V_G (see Response 1-1 for the relationship between V_G and ΔE_F), Fig. R4 shows that $\Delta\theta_{\mathbf{k}}$ appears only in line with the WTe_2 hinges. We also observed a clear sign flip of $\Delta\theta_{\mathbf{k}}$ when E_F is swept across the Dirac point of the hinge states. Third, $\Delta\theta_{\mathbf{k}}$ under the magnetic field (Fig. R4b) shows the gap opening of the hinge states. Under the external magnetic field B_z of 1 T, the localized $\Delta\theta_{\mathbf{k}}$ at the y -position of the hinges disappears when the Fermi level is close to the Dirac point (i.e., V_G near 0.88 V in the case of device #4), demonstrating the lifted degeneracy of hinge eigenstates due to the broken time-reversal symmetry. To summarize, the V_G - and B_z -dependent $\Delta\theta_{\mathbf{k}}$ distribution in device #4 is essentially identical to the devices without the graphene gap (see Fig. 2 and Figs. S4, 11 in the revised manuscript). We believe these data provide evidence that SHE is not likely the origin of our observation.

Figure R4. The spatially resolved V_G -dependent $\Delta\theta_k$ measured in device #4 when $B_z = 0$ (a) and $B_z = 1$ T (b). Dashed lines indicate the edge of each graphene layer, and the black rectangle represents the multilayer WTe_2 . The direction of the applied electric field is longitudinal along the $-x$ -direction, which is parallel to the WTe_2 a -axis.

We experimentally show that SHE is not the major origin of $\Delta\theta_k$ distribution. Generally speaking, the anomalous Hall effect (AHE) arises from the bulk or the surface due to the Berry curvature or scattering under spin-orbit coupled magnetization [Rev. Mod. Phys. **82**, 1539 (2010)]. It generates the transport of electrons transverse to the applied electric field, and more specifically, the transverse spin current is induced in the case of SHE. If SHE in the multilayer WTe₂ is the alternative origin of the observed $\Delta\theta_k$, then $\Delta\theta_k$ distribution should appear as Fig. R3c. However, the data shown in Fig. R4 indicate that this is not the case.

The way how the Berry curvature dipole affects the transverse electrical transport is different from SHE. Recent studies on the highly asymmetric WTe₂ revealed that the Berry curvature dipole is related to the nonlinear Hall effect in a non-magnetic environment [Nature **565**, 337-342 (2019); Nat. Mater. **18**, 324-328 (2019); Nat. Rev. Phys. **3**, 744-752 (2021)]. The nonlinear Hall current $\vec{j}^{2\omega}$ can be written as

$$\vec{j}^{\omega} = \frac{e^3\tau}{2(1+i\omega\tau)} \hat{z} \times \vec{E} (\vec{A} \cdot \vec{E}),$$

where ω is the frequency, τ is the Boltzmann transport relaxation time, and A is the Berry curvature dipole [Phys. Rev. Lett. **115**, 216806 (2015)]. Here, the nonlinear Hall effect induced by the Berry curvature dipole is nonlinear with respect to the applied sinusoidal electric field, and an electrical Hall effect causes a transverse charge current rather than the transverse spin current. Hence, $\Delta\theta_k$ distributions observed in our experiments are not related to the nonlinear Hall effect by the Berry curvature dipole, even though the anisotropic $\Delta\theta_k$ distribution matches that of the nonlinear Hall current. Moreover, even if the spin degree of freedom is assigned to the nonlinear Hall current for some reason, the nonlinear Hall effect should arise from the whole WTe₂ area, not exclusively from the hinge.

We have added the above discussions on the possible origins of $\Delta\theta_k$ other than the hinge states into the revised manuscript. The discussions on the results from a newly fabricated device (device #4) are provided in the revised main text (see Fig. 4 and the related discussions), and the further details about the experiments using device #4 are added in Supplementary Note 4 in the revised manuscript. The previous discussions on the orientation- and temperature-dependent $\Delta\theta_k$ (Fig. 4 in the original version of the manuscript) are moved to the Supplementary Information (see Supplementary Notes 3-2 and 3-3).

Comments 1-3: 3. Figure 2 (b) shows the spatially-resolved Kerr rotation. At a gate voltage of 0eV, especially 1eV, the signal is distributed over a large area (red area between -2.5 μm —2.5 μm), which is not consistent with the distribution of possible hinge states along the a-axis (near the top and bottom edges of the black rectangles). In addition, why did the authors choose a line cut with $x=0.75 \mu\text{m}$? This is some distance away from the area of possible hinge states.

Response 1-3: We thank Reviewer 1 for this important question regarding the experimental details. The contour plots in Fig. 2b show $\Delta\theta_k$ obtained by subtracting the Kerr rotation θ_k with and without the bias voltage between contact 1 and 3 (see Fig. 1 for the contact number). Note that the applied longitudinal electric field due to the bias voltage determines whether there is a field-induced electron injection from WTe_2 to graphene or not. In other words, the spatial $\Delta\theta_k$ distribution in our experiments depends on the spin transportation in graphene injected from WTe_2 , not the electrons remaining in the hinges.

There are the following technical reasons why we used graphene to detect the signature of the hinge states. First, the spatial length scale of the hinge states, whose width is only about a few nanometers [Nano Lett. **20**, 4228-4233 (2020); Nat. Phys. **14**, 918-924 (2018)], is far smaller than the spot size of the incident laser beam (diameter $\sim 1.5 \mu\text{m}$). This means that any optical signals originating from the hinge states would be extremely weak compared to that from the bulk, surface, or other background signals generated within the beam spot. This scaling problem limits the signal-to-noise ratio and hinders the direct detection of the hinge state if $\Delta\theta_k$ were to be measured on the laser-excited hinges. Secondly, because the hinge, by definition, is a junction between two surfaces, an incident laser introduces both the in-plane and the out-of-plane excitation at the two surfaces facing the hinge. In addition, while the light excitation occurs on a 2D surface, the 1D hinge may induce an anisotropic polarization-dependent reflection of light, thereby preventing the clear isolation of $\Delta\theta_k$ from the possible spin-polarized state.

To minimize the geometric effects of the hinge described above, we take advantage of graphene. Graphene is known to exhibit a long spin diffusion length of up to $30 \mu\text{m}$ due to the weak spin-orbit coupling and high mobility [Nano Lett. **16**, 3533-3539 (2016); Nat. Commun. **6**, 6766 (2015); Nat. Nanotech. **9**, 794-807 (2014)]. Under the electric field applied to the device, the electric potential gradient generates the longitudinal electron transport that injects electrons from WTe_2 to graphene, as described in Supplementary Note 2. The spin polarization of the injected electrons can survive long enough to be detected by optical measurements due to the long spin diffusion length of graphene. Therefore, because $\Delta\theta_k$ is distributed into the graphene

rather than the localized WTe₂ hinges, we have chosen a line cut at $x = 0.75 \mu\text{m}$ to analyze $\Delta\theta_k$ while keeping a certain distance from the edge of the WTe₂ crystal.

In the revised manuscript, we provided further details on the above discussion in Supplementary Note 2-1.

Comments 1-4: In addition to the above general comments, there are some suggestions

1. The thickness of the multilayer Td-WTe₂ used in the manuscript is 4nm (8 layer). In order to distinguish different possible topological phases in Td-WTe₂, such as Weyl semimetal, HOTI and 2D TI from 3D to 2D. The data of samples of different thickness are needed.

2. In the caption of Fig. 2, “V_G = -1.5, -1.2, 3 V are displayed” should be “V_G = -1.5, -1, 2, 3 V are displayed”.

In summary, I cannot be persuaded to recommend the manuscript in Nat. Commun.

Response 1-4: First of all, we apologize for not providing the correct information on the thickness of the sample. While preparing our responses, we found that the number of layers for the 4 nm thick T_d-WTe₂ is five rather than eight.

Along with Reviewer 1’s comments on the data from samples with different thicknesses, please note that Reviewer 2 also advised us to provide more data from additional devices. To address both Reviewers’ concerns, as well as to perform a self-consistency check, we fabricated new devices that have multilayer WTe₂ of various thicknesses. The optical microscope images of all devices used in our study are shown in Fig. R4. Figure R5 shows the thickness of WTe₂ in each device checked by the atomic force microscopy (AFM) measurements and the corresponding number of layers. Figure R6 shows the V_G-dependent electrical characteristics of the devices. The device number and the associated parameters (thickness, number of layers, and charge-neutral V_G, V_{CNP}) are summarized in Table R1. Please note that we have acknowledged two recent studies by K. Kang *et al.* (Nat. Mater. (2019)) and Z. Fei *et al.* (Nature (2018)) to estimate the number of layers from the AFM measurements. For the devices, device #1 is the one used to obtain Figs. 1-3 of the revised manuscript. Devices #2 and # 3 are the newly fabricated devices while preparing the response. Device #4 is also a new device with a slightly different structure, where there is a “spatial gap” in graphene (see Fig. R3a for the schematics) used to investigate the exclusive role of the hinge states on the $\Delta\theta_k$ distribution (see Response 1-2 for details).

Device	WTe ₂ thickness (nm)	Number of layers	V _{CNP} (V)
device #1	4	5	0.95
device #2	16	~20	1
device #3	3	4	0.92
device #4	5	7	0.88

Table R1. Thickness and V_{CNP} of each device used in the experiments. The thickness was measured by AFM (Fig. R5), and V_{CNP} was obtained from the V_{G} -dependent electrical transport (Fig. R6).

Figure R4. Optical microscopy images of devices #1-4. **a.** Device #1 is used to obtain the data in Figs. 1-3 in the revised manuscript. **b, c.** Devices #2 and #3 are newly fabricated devices with the same device structure as device #1. **d.** Device #4 has a 1.5 μm wide gap in the middle of graphene. All electrodes in the devices were made using 5 nm Ti and 25 nm Au. Numbers 1~4 in the figures are the contact indices. The black arrows in each figure show the orientation of WTe₂ crystal axes.

Figure R5. a-d. Thickness of the multilayer WTe_2 flake in the devices measured by AFM. The thickness of the metal contact (Ti 5 nm, Au 25 nm) close to WTe_2 is also measured for reference.

Figure R6. **a-b** V_G -dependent electrical characteristics obtained from devices #1-4. Contact 1-3 and 2-4 stand for the source and drain contact for each device. The current between contact 1, 3 (2, 4) corresponds to the current along the a - (b -) axis of each device.

Figure R7 shows the spatially resolved $\Delta\theta_k$ of devices #2 (Fig. R7a) and #3 (Fig. R7b) with varying V_G across the Dirac point (see Fig. R6 for the V_G -dependent transfer characteristics for each device). For device #1, the measured data are shown in Figs. 2 and 3 in the revised manuscript and also in Figs. S4-9 in Supplementary Information of the revised manuscript. For device #4, the associated contents are discussed in Response 1-2, and they are included in Fig. 4 in the revised manuscript. All the measurements were performed under the same condition as in the experiments performed with device #1 (e.g., cryogenic condition, chopping frequency, pump energy and power, bias voltage). From Fig. R7, we see that $\Delta\theta_k$ is localized at y -positions of the WTe₂ hinges, and its sign is flipped as V_G passes across V_{CNP} . These features are identical to the V_G -dependent distribution of $\Delta\theta_k$ observed in device #1.

Figure R7. Spatially resolved V_G -dependent $\Delta\theta_k$ from device #2 (a) and device #3 (b).

Let us provide more details on the thickness-dependent characteristics. First, the multilayer WTe_2 has been considered to be a type-II Weyl semimetal (WSM) [Nature **527**, 495-498 (2015)]. The characteristic features of WSM are the existence of the Weyl point pair and the

associated Fermi arcs connecting the two Weyl points. Investigation using surface-sensitive probes such as angle-resolved photoemission spectroscopy (ARPES) or scanning tunneling microscopy is a method to prove the WSM nature [Phys. Rev. B **94**, 121112(R) (2016); Phys. Rev. B **94**, 195134 (2016)]. Another is to see the “chiral anomaly”, known as the Adler-Bell-Jackiw anomaly in high-energy physics [Phys. Rev. **177**, 2426 (1969); Nuovo Cimento A **60**, 47-61 (1969)], by performing magneto transport experiments under parallel electric- and magnetic-field excitation, i.e., applying $\mathbf{E} \cdot \mathbf{B}$ [Science **350**, 6259 (2015), Phys. Rev. Lett. **126**, 185303 (2021), Nat. Rev. Phys. **3**, 394-404 (2021)]. Others include measurements of the quantum critical resistivity, showing a temperature-dependent inverse power law of resistivity [Phys. Rev. B **99**, 201112(R) (2019)]; though this is not a direct proof of the WSM nature. Unlike TaAs, TaP, and NbAs, it has not been clearly resolved the Weyl points and the Fermi arc in WTe₂, and the experimental evidence so far is not quite conclusive [Nano. Lett. **20**, 4228-4233 (2020); Nat. Mater. **19**, 974-979 (2020); Sci. Adv. **1**, e1501092 (2015); Nat. Commun. **11**, 1259 (2020)]. In our study, $\Delta\theta_{\mathbf{k}}$ near the hinge-localized states are not related to the above signatures of WSM, while the measurements show the conjectures of spinful HOTI as predicted in theoretical works and recent experiments in other HOTI materials [Phys. Rev. Lett. **123**, 186401 (2019); N. Shumiya *et al.*, Nat. Mater. (2022); Nat. Mater. **20**, 473-479 (2021)].

Second, in the monolayer limit of 1T'-WTe₂, the 2D quantum spin Hall insulator (QSHI) is now well established [Nat. Phys. **13**, 677-682 (2017); Nat. Phys. **13**, 683-687 (2017)]. Further studies reveal that a bilayer WTe₂ is topologically trivial and centrosymmetric [Adv. Mater. **28**, 4845-4851 (2016), Nano Lett. **22**, 5674-5680 (2022); Nature **565**, 337-342 (2019)]. Because our sample contains more than three layers, the multilayer WTe₂ is no longer considered to be a 2D system. If our sample were to exhibit a 3D TI phase, then the surface should host topologically protected gapless states. Searching for existing ARPES literature, we were not able to find any such evidence [Nat. Commun. **7**, 10847 (2016); Phys. Rev. B **94**, 121112(R) (2016)].

Lastly, a lower threshold of the layer numbers has been reported for T_d-WTe₂ to be a 3D HOTI [Nat. Mater. **19**, 974-979 (2020); Phys. Rev. Lett. **123**, 186401 (2019)], which dictates that it should be at least three, while no theoretical or experimental study has reported the upper limit of the thickness of T_d-WTe₂ that guarantees the existence of the HOTI phase. For example, a recent study has shown that even a 20-nm-thick WTe₂ (~ 25 atomic layers) can host a conducting channel localized at the hinge [Nat. Mater. **19**, 974-979 (2020)]. We expect that further theoretical studies are required to perform to determine the exact crossover of layer numbers and upper bound for HOTI to occur.

In the revised manuscript, we have included a more detailed discussion. Table R1 and Figs. R4-7 are added to the Supplementary Information as Table S1 and Figs. S2-4 and Fig. S13, respectively. Corresponding contents on the data from devices with various thicknesses are added in Supplementary Note 3-4. We have also edited the caption of Fig. 2 as commented by Reviewer 1.

General remarks and comments of Reviewer 2: The experimental study reported in this manuscript is a well-motivated and timely one, aimed at finding experimental evidence for the helical Dirac nature of higher order topological insulator hinge states that are predicted to exist in Td-WTe₂. This is a challenging and ambitious goal since the usual approach of spin- and angle- resolved photoemission does not have the spatial and energy resolution to probe the spin polarization of these states. This manuscript reports a clever idea for observing the putative spinful hinge states via spatially-resolved polar magneto-optic Kerr effect (MOKE) measurements on very thin WTe₂-graphene heterostructure devices and claims that the observations provide evidence for spinful hinge states. The experiment is carried out in the following geometry: a flake of WTe₂ is placed in the middle of a larger graphene device that carries a current along either the a axis or the b axis. The entire device can be back gated, allowing the Fermi energy to be varied in both the graphene and WTe₂ layers. In the region of the WTe₂ flake, this current flows in parallel through the graphene and WTe₂ channels. If spinful hinge states exist, this should result in a spin accumulation at the edges of the WTe₂ and create a spin polarization that should diffuse into the vicinal graphene. Spatially-resolved MOKE measurements then probe the spin polarization in the device using the Kerr rotation. This shows that when the current flows along the a axis, a spin polarized signal is observed via MOKE in the graphene near the edges of the WTe₂ flake. A key feature here is that the MOKE signal is strongly dependent on the back gate voltage (V_g) and changes sign as the Fermi energy is swept through the charge neutral point of WTe₂. The signals are quite small but the signal-to-noise is good enough to produce reasonably convincing signatures of spin polarization. The authors also study the MOKE signal as a function of magnetic field and argue that its disappearance at higher fields is consistent with the requirement of time reversal symmetry for the presence of spinful hinge states. The authors do a reasonably systematic job of ruling out some obvious explanations (e.g. Landau levels, quantum Hall effect).

I find all these observations to be very intriguing, interesting, and puzzling. It's an elegant experiment, ambitious in its goal, and it would be worthwhile publishing this data in a high profile journal such as Nature Communications. However, the manuscript in its current form has some major deficiencies.

Response: We appreciate Reviewer 2's time, consideration, and detailed comments. Indeed, theoretical investigations on the higher-order topological insulator (HOTI) have been predicted in many different contexts: first-principle calculation [Phys. Rev. Lett. **123**, 186401 (2019)], topological field theory [Phys. Rev. B **98**, 235102 (2018)], and density functional theory [Phys. Rev. Lett. **123**, 216803 (2019); Phys. Rev. Research **3**, L042044 (2021)]. On the experiment side, investigations including supercurrent interferometry and scanning

tunneling microscopy have shown evidence about the gapless hinges [Nat. Mater. **19**, 974-979 (2020); Nat. Commun. **12**, 4420 (2021)]. Because of the spatially localized nature of hinges between two gapped surface states, it is highly nontrivial to probe the hinge states directly. Furthermore, the hinges, if they are present, are sources of charged carrier scattering, providing additional complications. In this respect, we are very much grateful to Reviewer 2 for acknowledging our approach of using graphene/WTe₂ heterodevices.

As one of our major efforts in preparing the responses, we fabricated a few more devices (device #2 - device #4) and checked the reproducibility of experiments with all the devices (Comment 2-3). An additional device (device #4) was fabricated with a slightly different geometry. This was used to demonstrate that the spin Hall effect (SHE) is not likely the origin of our observed $\Delta\theta_k$ distribution (Comment 2-1).

Please note that the Extended data and Methods in the original manuscript are moved to Supplementary Information in the revised manuscript. These include a more detailed explanation of the experimental design and the results from additional experiments performed with devices #2-4.

Below we present our point-by-point response to Reviewer 2's comments.

Comments 2-1: I note that in principle, just a regular spin Hall effect in WTe₂ could also produce such signals by sourcing spin current into the graphene. I believe that this would also change sign as the Fermi energy passes through the charge neutral point. The authors carry out a measurement with the current flowing along the b axis to rule this out, showing that a localized spin polarization is not observed near the edges. However, as the authors correctly note, the spin-orbit coupling is also much weaker for this case. So, I am not convinced that one can rule out the conventional spin Hall effect and immediately jump to the conclusion that the observed MOKE signal is arising from spinful hinge states in WTe₂.

Response 2-1: We agree with Reviewer 2's comments that it is highly desirable to experimentally exclude other possible explanations, especially SHE. In fact, Reviewer 1 and Reviewer 3 raised a similar concern (Comment 1-2, Comment 3-1) and advised us to provide additional proof to make our claim solid. Because Reviewer 1 and Reviewer 3 raised a similar concern, we may reuse our response to such a similar comment, as shown below.

It is essential to show how SHE the $\Delta\theta_{\mathbf{k}}$ distribution and prove that only the helical hinge contributes to $\Delta\theta_{\mathbf{k}}$. In general, the intrinsic and extrinsic contribution of SHE arise due to the Berry curvature and scattering from the spin-orbit coupled magnetization, respectively [Rev. Mod. Phys. **82**, 1539 (2010)]. Under a longitudinal electric field, such an effect invokes a transverse electronic transport of spin-polarized electrons. Thus, the spinful version of anomalous Hall effect (i.e., SHE) generates a transverse spin current. In this light, there is a possibility that SHE induced in the multilayer WTe₂ might be an alternative origin of our observation.

To address this issue, we first wish to show how we exclude SHE experimentally from the source of $\Delta\theta_{\mathbf{k}}$ using the data shown in Fig. 2 in the revised manuscript. If the localized $\Delta\theta_{\mathbf{k}}$ arises from SHE, the spin-polarized electrons would be spatially distributed more along the transverse direction to the applied electric field (along y-direction) (see Fig. R8a) rather than along the longitudinal direction (along x-direction), i.e., parallel to the electric field. However, data shown in Fig. 2b in the revised manuscript clearly show that $\Delta\theta_{\mathbf{k}}$ is an elliptical shape whose major distribution axis is more inclined parallel to the applied electric field direction.

Figure R8. Schematic diagrams representing the expected $\Delta\theta_k$ distribution when the spin-polarized electrons detected in the graphene are injected from the bulk of WTe_2 , i.e., due to SHE (**a**), and when they are injected from the hinges of WTe_2 (**b**). The dashed rectangles are the spatial window that we performed the Kerr rotation measurements. Red, blue and black arrows represent the spin-polarized electronic transport of two opposite spins, respectively.

To provide additional evidence on the above discussion as well as to address Reviewer's concern, we have fabricated a new device (device #4) having almost the same structure but with a $1.5\ \mu\text{m}$ gap in the middle of graphene. The optical microscopy image of the device is shown in Fig. R9a. If $\Delta\theta_k$ originates from the hinge states, the localized $\Delta\theta_k$ should arise only at the y -position of the WTe_2 hinge (Fig. R9b). On the other hand, if the observed $\Delta\theta_k$ originates from the bulk spin transport in WTe_2 (i.e., due to SHE), the spin-polarized electrons injected into graphene are expected to be spread out to the left as well as to the right of the graphene area in a transverse direction to the applied electric field (Fig. R9c). Therefore, the presence of a gap in graphene would collect the accumulated spin-polarized electrons at the edge of graphene on both sides of the gap (Fig. R9c).

Figure R9. An experiment proposed to isolate the hinge characteristics (device #4). **a** An optical microscopy image is shown. Two monolayer graphene flakes are separated by a $1.5 \mu\text{m}$ gap. This device scheme is almost identical to the other devices, except for the presence of a gap in graphene. The graphene layer for the electron transport measurement is located below WTe_2 . **b, c** Schematic representation for the expected $\Delta\theta_{\mathbf{k}}$ distribution when the spin-polarized electrons in graphene originate from the WTe_2 hinge states (**b**) and when they originate from the WTe_2 bulk (**c**). Dashed rectangles are the spatial windows that we performed the Kerr-rotation measurements. The black arrows are to represent electron transport.

To check the above idea, we have investigated the magneto-optic Kerr effect on device #4. The results are shown in Fig. R10. Here we performed the measurements of the spatially resolved $\Delta\theta_{\mathbf{k}}$ at different V_G with (Fig. R10a) and without (Fig. R10b) an external magnetic field ($B_z = 1$ T). We first note that no accumulation of the spin-polarized electrons was seen on either side of the graphene gap, regardless of V_G . Secondly, with varying V_G (see Response 2-3 for the relationship between V_G and ΔE_F), Fig. R10a shows that $\Delta\theta_{\mathbf{k}}$ appears only in line with the WTe_2 hinges. We also observed a clear sign flip of $\Delta\theta_{\mathbf{k}}$ when E_F is crossing the Dirac point of the hinge states. Third, the $\Delta\theta_{\mathbf{k}}$ under magnetic field (Fig. R10b) shows the gap opening of the hinge states. Under the external magnetic field B_z of 1 T, the localized $\Delta\theta_{\mathbf{k}}$ at the y -position of the hinges disappears when the Fermi level is close to the Dirac point (i.e., V_G is near 0.88 V in the case of device #4), demonstrating the lifted degeneracy of hinge eigenstates due to the broken time-reversal symmetry. To summarize, the V_G - and B_z -dependent $\Delta\theta_{\mathbf{k}}$ distribution in device #4 is essentially identical to the devices without the graphene gap (see Fig. 2 and Figs. S4, 11 in the revised manuscript). We believe these data provide evidence that SHE is not likely the origin of our observation.

Figure R10. The spatially resolved V_G -dependent $\Delta\theta_k$ measured in device #4 when $B_z = 0$ (a) and $B_z = 1$ T (b). Dashed lines indicate the edge of each graphene layer, and the black rectangle represents the multilayer WTe_2 . The direction of the applied electric field is longitudinal along the $-x$ -direction, which is parallel to the WTe_2 a -axis.

We experimentally show that SHE is not the major origin of $\Delta\theta_{\mathbf{k}}$ distribution. Generally speaking, the anomalous Hall effect arises from the bulk or the surface due to the Berry curvature or scattering under spin-orbit coupled magnetization [Rev. Mod. Phys. **82**, 1539 (2010)]. It generates the transport of electrons transverse to the applied electric field, and more specifically, the transverse spin current is induced in the case of SHE. If SHE in the multilayer WTe₂ is the alternative origin of the observed $\Delta\theta_{\mathbf{k}}$, then $\Delta\theta_{\mathbf{k}}$ distribution should appear as Fig. R3c. However, the data shown in Fig. R4 indicate that this is not the case.

The way how the Berry curvature dipole affects the transverse electrical transport is different from SHE. Recent studies on the highly asymmetric WTe₂ revealed that the Berry curvature dipole is related to the nonlinear Hall effect in a non-magnetic environment [Nature **565**, 337-342 (2019); Nat. Mater. **18**, 324-328 (2019); Nat. Rev. Phys. **3**, 744-752 (2021)]. The nonlinear Hall current $\vec{j}^{2\omega}$ can be written as

$$\vec{j}^{\omega} = \frac{e^3\tau}{2(1+i\omega\tau)} \hat{z} \times \vec{E} (\vec{A} \cdot \vec{E}),$$

where ω is the frequency, τ is the Boltzmann transport relaxation time, and A is the Berry curvature dipole [Phys. Rev. Lett. **115**, 216806 (2015)]. Here, the nonlinear Hall effect induced by the Berry curvature dipole is nonlinear with respect to the applied sinusoidal field, and an electrical Hall effect causes a transverse charge current rather than the transverse spin current. Hence, the $\Delta\theta_{\mathbf{k}}$ distributions observed in our experiments are not related to the nonlinear Hall effect by the Berry curvature dipole, even though the anisotropic $\Delta\theta_{\mathbf{k}}$ distribution matches that of the nonlinear Hall current. Moreover, even if the spin degree of freedom is assigned to the nonlinear Hall current for some reason, the nonlinear Hall effect should arise from the whole WTe₂ area, not exclusively from the hinge.

We have added the above discussions on the possible origins of $\Delta\theta_{\mathbf{k}}$ other than the hinge states into the revised manuscript. The discussions on the results from a newly fabricated device (device #4) are provided in the revised main text (see Fig. 4 and the related discussions), and further details about the experiments using device #4 are added in Supplementary Note 4 in the revised manuscript. The previous discussions on the orientation- and temperature-dependent $\Delta\theta_{\mathbf{k}}$ (Fig. 4 in the original version of the manuscript) are moved to the Supplementary Information (see Supplementary Notes 3-2 and 3-3).

Comments 2-2: (I do not understand the data shown as "hinge" and "bulk" in Fig. 4b -- the paper does not explain whether this is taken for current flow along the a axis or b axis -- I guess it is the former but it's confusing since Fig. 4a focuses on current flowing along the b axis.)

Response 2-2: We feel sorry for the confusion in reading the discussion in Fig. 4. The figure contains two different experiments, and those data should not have been packed in one figure. Below, we first provide more details on the discussion in Fig. 4 and then present how we modify the associated contents to avoid such confusion.

The contour plots in Fig. 4a in the original manuscript were spatially resolved $\Delta\theta_{\mathbf{k}}$ from the V_G -dependent $\Delta\theta_{\mathbf{k}}$ measurement with the bias voltage applied along the b -axis. Unlike the spinful hinges along the a -axis, measurements along the b -axis did not show the spinful characteristics of the hinge states. Figure 4b in the original manuscript compares $|\Delta\theta_{\mathbf{k}}|_{\text{avg}}$ obtained near the hinge and bulk as a function of temperature. There, we have seen that $\Delta\theta_{\mathbf{k}}$ from the hinge and bulk exhibits a different temperature dependence, based on which we assumed that the localized $\Delta\theta_{\mathbf{k}}$ near the hinges is not simple confinement of spin-polarized electrons from the bulk-originated effects. While those results and interpretations in the original manuscript were meant to provide supporting evidence for the spinful HOTI characteristics, we realized that those ideas do not provide direct evidence. Due to this reason, we have moved them into Supplementary Notes 3-2 and 3-3 in the revised manuscript. Instead, the contents of Fig. 4 are replaced with the discussions and results obtained from additional experiments using newly fabricated device #4 (see Response 2-1, Figs. R9, 10). We believe that this offers more convincing arguments in excluding other contributions on the spinful hinge states.

Comments 2-3: First, I always find it troublesome to see an experimental manuscript that only shows data from a single device. It's crucial to show that the observations can be reproduced and give the reader some idea of how readily this experiment can be repeated. Was this one device out of many in which these observations could be seen? Or do the authors have data on additional devices? Why not show all the data in the supplementary section and make a comment on the statistical device-to-device variations?

Response 2-3: We completely agree with Reviewer 2 that it is worthwhile to check the reproducibility. As we have made a statement to the general comment of Reviewer 2, we have fabricated a total of three more devices. Two additional devices (device #2 and #3) have essentially the same geometry as device #1 except for the WTe₂ thickness. For device #4, we have used it to investigate SHE contribution and see if we can isolate the spinful hinge characteristics (Response 1-2, 2-1, 3-1). The details of the samples used in our experiments are discussed below.

The optical microscope images of all devices used in our study are shown in Fig. R11. Figure R12 shows the thickness of WTe₂ in each device checked by the AFM measurements and the corresponding number of layers. Figure R13 shows the V_G -dependent electrical characteristics of the device. The device number and the associated parameters (thickness, number of layers, and charge-neutral V_G , V_{CNP}) are summarized in Table R2. Please note that we have acknowledged two recent studies by K. Kang *et al.* (Nat. Mater. (2019)) and Z. Fei *et al.* (Nature (2018)) to estimate the number of layers from the AFM measurements. For the devices, device #1 is the one used to obtain Figs. 1-3 of the revised manuscript. Devices #2 and #3 are the newly fabricated devices while preparing the response. Device #4 is also a new device with a slightly different structure, where there is a “spatial gap” in graphene (see Fig. R3a for the schematics) used to investigate the exclusive role of the hinge states on the $\Delta\theta_K$ distribution (see Response 2-1 for details).

Device	WTe ₂ thickness (nm)	Number of layers	V_{CNP} (V)
device #1	4	5	0.95
device #2	16	~20	1
device #3	3	4	0.92
device #4	5	7	0.88

Table R2. Thickness and V_{CNP} of each device used in the experiments. The thickness was measured by AFM (Fig. R12), and V_{CNP} was obtained from the V_G -dependent electrical transport (Fig. R13).

Figure R11. Optical microscopy images of devices #1-4. **a.** Device #1 is used to obtain the data in Figs. 1-3 in the revised manuscript. **b, c.** Devices #2 and #3 are newly fabricated devices with the same device structure as device #1. **d.** Device #4 has a 1.5 μm wide gap in the middle of graphene. All electrodes in the devices were made using 5 nm Ti and 25 nm Au. Numbers 1-4 in the figures are the contact indices. The black arrows in each figure show the orientation of WTe_2 crystal axes.

Figure R12. a-d. Thickness of the multilayer WTe_2 flake in the devices measured by AFM. The thickness of the metal contact (Ti 5 nm, Au 25 nm) close to WTe_2 is also measured for reference.

Figure R13. a-d V_G -dependent electrical characteristics obtained from devices #1-4. Contact 1-3 and 2-4 stand for the source and drain contact for each device. The current between contact 1, 3 (2, 4) corresponds to the current along the a - (b -) axis of each device.

Figure R14 shows the spatially resolved $\Delta\theta_{\mathbf{k}}$ of devices #2 (Fig. R14a) and #3 (Fig. R14b) with varying V_G across the Dirac point (see Fig. R13 for the V_G -dependent transfer characteristics for each device). For device #1, the measured data are shown in Figs. 2 and 3 in the revised manuscript and also in Figs. S4-9 in Supplementary Information of the revised manuscript. For device #4, the associated contents are discussed in Response 2-1, and they are included in Fig. 4 in the revised manuscript. All the measurements were performed under the same condition as in the experiments performed with device #1 (e.g., cryogenic condition, chopping frequency, pump energy and power, bias voltage). From Fig. R14, we see that $\Delta\theta_{\mathbf{k}}$ is localized at y -positions of the WTe_2 hinges, and its sign is flipped as V_G passes across V_{CNP} WTe_2 . These features are identical to the V_G -dependent distribution of $\Delta\theta_{\mathbf{k}}$ observed in device #1.

Figure R14. Spatially resolved V_G -dependent $\Delta\theta_k$ from device #2 (a) and device #3 (b).

Let us provide some discussion on the thickness-dependent characteristics. First, multilayer WTe_2 has been considered to be a type-II Weyl semimetal (WSM) [Nature **527**, 495-498 (2015)]. The characteristic features of WSM are the existence of the Weyl point pair and the associated Fermi arcs connecting the two Weyl points. Investigation using surface-sensitive probes such as angle-resolved photoemission spectroscopy (ARPES) or scanning tunneling

microscopy is one method to prove the WSM nature [Phys. Rev. B **94**, 121112(R) (2016); Phys. Rev. B **94**, 195134 (2016)]. Another is to see the “chiral anomaly”, known as the Adler-Bell-Jackiw anomaly in high-energy physics [Phys. Rev. **177**, 2426 (1969); Nuovo Cimento A **60**, 47-61 (1969)], by performing magneto transport experiments under parallel electric- and magnetic-field excitation, i.e., applying $\mathbf{E} \cdot \mathbf{B}$ [Science **350**, 6259 (2015); Phys. Rev. Lett. **126**, 185303 (2021); Nat. Rev. Phys. **3**, 394-404 (2021)]. Others include measurement of the quantum critical resistivity showing a temperature-dependent inverse power law of resistivity [Phys. Rev. B **99**, 201112(R) (2019)], though this is not a direct proof of WSM nature. Unlike TaAs, TaP, and NbAs, it has not been clearly resolved the Weyl points and the Fermi arc in WTe₂, and experimental evidence so far is not quite conclusive [Nano. Lett. **20**, 4228-4233 (2020); Nat. Mater. **19**, 974-979 (2020); Sci. Adv. **1**, e1501092 (2015); Nat. Commun. **11**, 1259 (2020)]. In our study, $\Delta\theta_{\mathbf{k}}$ near the hinge-localized states are not related to the above signatures of the WSM, while the measurements show the conjectures of spinful HOTI as predicted in theoretical works and observed in other HOTI materials [Phys. Rev. Lett. **123**, 186401 (2019); N. Shumiya *et al.*, Nat. Mater. (2022); Nat. Mater. **20**, 473-479 (2021)].

Second, in the monolayer limit of 1T'-WTe₂, the 2D quantum spin Hall insulator (QSHI) is now well established [Nat. Phys. **13**, 677-682 (2017); Nat. Phys. **13**, 683-687 (2017)]. Further studies reveal that a bilayer WTe₂ is topologically trivial and centrosymmetric [Adv. Mater. **28**, 4845-4851 (2016); Nano Lett. **22**, 5674-5680 (2022); Nature **565**, 337-342 (2019)]. Because our sample contains more than three layers, the multilayer WTe₂ is no longer considered to be a 2D system. If our sample were to exhibit a 3D TI phase, then the surface should host topologically protected gapless states. Searching for existing ARPES literature, we were not able to find any such evidence [Nat. Commun. **7**, 10847 (2016); Phys. Rev. B **94**, 121112(R) (2016)].

Lastly, a lower bound threshold of the layer numbers has been reported for T_d -WTe₂ to be a 3D HOTI [Nat. Mater. **19**, 974-979 (2020); Phys. Rev. Lett. **123**, 186401 (2019)], which dictates that it should be at least three, while no theoretical or experimental study has reported the upper limit of the thickness of T_d -WTe₂ that guarantees the existence of the HOTI phase. For example, a recent study has shown that even a 20-nm-thick WTe₂ (~ 25 atomic layers) can host a conducting channel localized at the hinge [Nat. Mater. **19**, 974-979 (2020)]. We expect that further theoretical studies are required to perform to determine the exact crossover of layer numbers and upper bound for the HOTI to occur.

In the revised manuscript, we have included a more detailed discussion. Table R2 and Figs. R11-14 are added to the Supplementary Information as Table S1 and Figs. S2-4 and Fig. S13,

respectively. Corresponding descriptions of the data from devices with various thickness is added in Supplementary Note 3-4.

Comments 2-4: Second, it is important to show some control experiment. For example, is there a way to make a device wherein the WTe₂ layer is not predicted to host spinful hinge states? Does the experiment show the complete absence of a signal in that case?

Response 2-4:

We fully agree that a control experiment is important to verify our main idea. The topological phase, including higher-order topology, is an intrinsic property of materials in which the topological nature cannot be changed unless the associated band topology is broken or the band structure is manipulated, e.g., breaking the time-reversal symmetry via a magnetic means. In our case, when the applied magnetic field is strong enough (~ 1 T), we have not seen the characteristics of HOTI. From a different point of view, the number of stacked WTe₂ layers might be a control parameter to manipulate the higher-order topology in terms of the band structure. If we can make a device with a multilayer WTe₂ with a specific number of layers exhibiting no higher-order band topology, it will be an ideal control experiment. Then, the remaining question is to check if there is a boundary of thickness “threshold” for the higher-order topological phase. For that purpose, we have fabricated multiple WTe₂/graphene devices with different thicknesses (device #2~4). The details of devices and resultant $\Delta\theta_k$ obtained from the devices were discussed in Responses 2-1 and 2-3. All the multilayer WTe₂ samples we used in the additional experiments were revealed to exhibit the helical hinge states as shown in Figs. R10 and R14.

When the layer number is reduced down to a mono- and bilayer limit, such cases might also be considered to be alternative candidates for the control experiments. Their topological phases are, however, already known to be different from that of the multilayer WTe₂. For the monolayer WTe₂ (1T'-WTe₂), it is well known that the material displays the helical 2D TI nature [Nat. Phys. **13**, 677-682 (2017); Nat. Phys. **13**, 683-687 (2017)]. For the bilayer WTe₂, it is topologically trivial, having a large Berry curvature dipole [Nature **565**, 337-342 (2019); Nano Lett. **22** 5674-5680 (2022)]. Here, further discussions are needed to be addressed. Please note that the monolayer WTe₂ has helical 1D edge states as 2D TI, which can be considered as a source of spin-polarized electrons localized at the 1D edge, just like the hinge states of multilayer WTe₂ [Sci. Adv. **5**, eaat8799 (2019); Nat. Commun. **8**, 659 (2017)]. In this case, if one completely isolates the edge (excluding the bulk), the electrons injected from the edge states of monolayer WTe₂ to graphene may cause the same spatial distribution of $\Delta\theta_k$, similar to the case of our experiments. For the bilayer WTe₂, no 1D localized state was reported both theoretically and experimentally [Adv. Mater. **28**, 4845-4851 (2016); Sci. Adv. **5**, eaat8799 (2019); Nano Lett. **22** 5674-5680 (2022)]. Especially, Y. Shi *et al.* (Sci. Adv. (2019)) and F. Lüpke *et al.* (Nano Lett. (2022)) have already provided explicit evidence of trivial edges in the bilayer WTe₂ using microwave impedance

microscopy and scanning tunneling spectroscopy. Although the bilayer WTe₂, unlike the multilayer WTe₂, has an inversion symmetry with a topologically distinct phase, it is the most similar system to a multilayer WTe₂ with no spinful hinge states. Therefore, measuring $\Delta\theta_{\mathbf{k}}$ with the device made of bilayer WTe₂ would be an appropriate demonstration as a control experiment. Although we were not able to identify an appropriate bilayer sample while preparing our revision, we would be willing to conduct additional experiments if Reviewer 2 further requests.

In the original manuscript, we have provided experiments with a varying magnetic field (B_z) and temperature as for the control experiments; temperature-dependent $\Delta\theta_{\mathbf{k}}$ distribution data are now moved to Fig. S9 in the Supplementary Information. There, we have shown that $\Delta\theta_{\mathbf{k}}$ distributions are disappeared when WTe₂ is no longer HOTI due to the applied B_z and at an elevated temperature. Here, the applied B_z breaks the time-reversal symmetry, and the increased temperature activates a strong electron-phonon interaction, both of which hinder observing the topological effects. Figure R15 summarizes the results of B_z - and temperature-dependent data. $\Delta\theta_{\mathbf{k}}$ from the hinges, in the case that V_G is close to the Dirac point, gradually disappears with increasing B_z , which arises due to the mass gap opening at the Dirac point (Fig. R15a-c). We have especially seen that $\Delta\theta_{\mathbf{k}}$ signals of the spinful hinges completely disappeared at a high temperature above 40 K (Fig. R15d), which is, in fact, consistent with the reported theoretical works [Phys. Rev. B **102**, 165153 (2020); Phys. Rev. Lett. **123**, 186401 (2019)].

In the revised manuscript, we have further provided the above discussions. The temperature dependence is discussed in Supplementary Note 3-3, and the B_z - dependence is discussed in Supplementary Note 3-2. The $\Delta\theta_{\mathbf{k}}$ data obtained from devices #2~4 is discussed in Supplementary Note 3-4.

Figure R15. a-c Line-cut plots obtained from the spatially resolved V_G -dependent $\Delta\theta_K$ with varying B_z . d Temperature dependence of the hinge and bulk contribution of $|\Delta\theta_K|_{\text{avg}}$.

Comments 2-5: Finally, while the paper is reasonably well-organized and easily followed by a specialist who is interested enough to ignore the superficial language issues, the paper could be more carefully edited to improve the readability for non-specialists. Below, I give some examples of sentences that are problematic.

Until all the deficiencies that I have outlined above are taken care of, especially measurements on additional devices and a control, even with the very interesting data obtained on the devices shown in the current manuscript, I cannot recommend that this paper be published in Nature Communications.

A few examples of awkward or grammatically incorrect language:

Abstract:

"Higher-order topological insulator is a recently discovered topological phase of matters with generalized bulk-boundary correspondence."

"Our finding may invoke stimuli to extend the previous spinless models and further provide a fertile diagnosis to the higher-order topology and Weyl physics in multidimensional solids."

Page 3: "The elemental origins of the gapped surface state are the..."

Page 3: "Among such candidates, WTe₂ has drawn crucial interests in the perspective of band 1 topology."

Page 6: "One seeming method to examine such topological protection..."

Page 7: "Without B_z, the hinge states host the gapless distribution with Dirac points protected by the time-reversal symmetry."

Page 7: "With increasing B_z ≥ 0.5 T, the oscillatory delta-Theta_K subdue in the broader range of VG."

Page 7: "...or WTe₂ exhibits no the HOTI phase."

Page 8: "...the new massless position cannot occur between two surfaces states, which ascertains the mass gap open in our experiments."

Response 2-5: We are grateful to Reviewer 2 for his/her careful examination and suggestion to improve the readability of our manuscript. We apologize for not being proficient in language and grammar. Following Reviewer 2's advice, we carefully checked the whole manuscript again and made corrections, including the points suggested by Reviewer 2. Below is the list of Edited contents.

Edited contents

Edited sentences

- Abstract

(page 1 line 15)

Higher-order topological insulators are recently discovered quantum materials exhibiting distinct topological phases with the generalized bulk-boundary correspondence.

(page 1 line 21)

Here, we employ the magneto-optic Kerr effect to visualize the spinful characteristics of the hinge states in a few-layer T_d -WTe₂.

(page 2 line 3)

Our experiment provides a fertile diagnosis to investigate the topologically protected gapless hinge states, and may call for new theoretical studies to extend the previous spinless model.

- Main text

(page 3, line 5)

Such a phenomenon can be understood based on the fact that the gapped surface states host a doubly inverted electronic band and the strong spin-orbit coupling (SOC).

(page 3, line 13)

Among such candidates, WTe₂ has recently attracted many interests to investigate the electronic correlations as well as to explore the topologically protected quantum phenomena.

(page 4, line 8)

To investigate the spin orientation of the hinge states,

(page 4, line 10)

Our results agree with the previous spin-resolved observations of WTe₂, ~

(page 4, line 14)

Furthermore, we examine the time-reversal invariance of the spinful hinge states by opening the mass gap via external magnetic fields.

(page 4, line 23)

~ the multilayer WTe₂, ~

(page 5, line 3)

with a micrometer-scale resolution.

(page 5, line 8)

the bulk and the hinge contribution.

(page 5, line 11)

where it is designed to perform the electrical and optical measurements along both axes.

(page 5, line 17)

The observed two conductance deeps at $V_G = 0.5$ and 0.95 V ~

(page 6, line 5)

~hinge-originated $\Delta\theta_k$ in details.

(page 6, line 7)

~ with increasing V_G , and $\Delta\theta_k$ changes ~

(page 6, line 14)

These results strongly suggest that although the spin configuration of helical hinge states of the bottom surface of multilayer WTe₂ resembles that of the spin-momentum-locked helical edge states of the 2D quantum spin Hall insulator, the multilayer WTe₂ is not simply a stack of weak 2D topological insulator layers as proven previously.

(page 6, line 19)

One method to examine such topological protection,

(page 6, line 23)

In Fig. 3a, where B_z is 0.5 T, we see that the oscillatory $\Delta\theta_k$ near the hinges vanishes as V_G approaches the charge neutrality. With increasing B_z of 1 T (Fig. 3b), the $\Delta\theta_k$ signals near the hinge are more suppressed compared to the case of B_z of 0.5 T, yet the localized $\Delta\theta_k$ survives only when V_G is pushed further below (0 V) and above (1.5 V) the charge neutrality point. When B_z is sufficiently large, Fig. 3c shows the

vanishing $\Delta\theta_{\mathbf{k}}$ signals, which imply that no spin-polarized electrons are present; this can be readily understood as the mass gap opening due to the broken time-reversal symmetry.

(page 7, line 8)

Without B_z , the hinge states remain gapless because the Dirac point is protected by the time-reversal symmetry.

(page 7, line 11)

Such disappearance of $\Delta\theta_{\mathbf{k}}$ characteristics when $B_z = 2$ T may ~

(page 7, line 13)

~ or WTe₂ exhibits no HOTI phases with increasing external magnetic fields.

(page 7, line 18)

~ opening.

(page 8, line 3)

In our experiment regime, the applied B_z makes the hinge being a boundary between one parallel to B_z and another perpendicular to B_z .

(page 8, line 5)

~such spatially shifted hinge modes cannot occur between the two surface states.

(page 8, line 7 ~ page 9, line 10)

The transverse spin accumulation ~ the origin of our observation.

(page 9, line 15)

The V_G - and B_z -dependent data provide strong evidence that the helical spin-polarized states are within the bulk bandgap while they are localized at the geometric hinge of the multilayer WTe₂, whose energy degeneracy is protected by the time-reversal symmetry.

(page 9, line 21)

~ in other higher-order topological materials.

- References 15 and 16 are added.

General remarks and comments of Reviewer 3:

- What are the noteworthy results?

The results of note are the claim to direct observation of hinge states in WTe₂.

- Will the work be of significance to the field and related fields? How does it compare to the established literature? If the work is not original, please provide relevant references.

According to the authors, there already exists evidence for hinge states from an experiment where WTe₂ was coupled to a superconductor. The claim here is that this experiment is a “direct observation of hinge states” and resolves the “spinful” nature of the transport. If correct, this would be a significant contribution. However, as I discuss below, I am not at all clear on the evidence supporting these conclusions.

Response: We appreciate Reviewer 3’s general assessment as well as a concern about the recent reports of Y.-B. Choi *et al.* (Nat. Mater. (2020)) and A. Kononov *et al.* (Nano Lett. (2020)). These studies have focused on the existence of the hinge states by analyzing the supercurrent interference of conducting edges. Both works have exploited the Josephson junction and have found that the hinges are strongly localized, whose conductivity anisotropy is consistent with the anisotropic geometry, i.e., *a*- and *b*-axis. One thing that we wish to mention is that such investigations rely on analyzing the supercurrent interference and the associated Fraunhofer pattern, in which a moderate amount of magnetic field should be applied, typically a few mT to 1 T, to observe a clear interference. We believe that this is why those studies were not able to uncover the spinful nature of the hinge channels.

In this respect, our motivation is to examine not only the presence of the hinges but also to probe the spin configuration of such higher-order hinge states. We have conducted spatially resolved optical measurements with varying electrostatic doping to isolate the gapless spinful hinge channel, as well as applying an external magnetic field to inspect the topological nature of the hinge states. In preparation for our responses, we have fabricated three more devices (devices #2, #3, and #4) and performed the same experiment as presented in the original manuscript. Below, we present more detailed explanations of how we experimentally characterized the spinful characteristics and how we were able to distinguish the bulk contribution (e.g., spin Hall effect).

Comments 3-1:

- Does the work support the conclusions and claims, or is additional evidence needed?

I have some questions about the fundamentals of the experiment that are not explained in the manuscript. The most basic is how the authors interpret the asymmetric spin accumulation that they observe. I understand the hinge states to be 1D channels at the surface. The spin polarization of the hinge states is not clear to me. Does each side of the surface support two counterpropagating spin up and down modes, as in the Quantum Spin Hall Effect? It would be very useful to clarify this point.

If this is indeed the case, it is not clear to me how to account for spin polarization on the basis of hinge states alone. The asymmetric spin accumulation presumably arises from a spin current transverse to the current provided by the voltage bias. Isn't it necessary that this current arise from the bulk in as much as the hinge states support only purely longitudinal currents?

Based on this, I would tend to argue that the bulk spin Hall Effect (SHE) must be involved in generating the spin accumulation that is observed. The authors appear to claim is that this effect is ruled out by two observations. The first is that the spin accumulation is not seen when the Fermi level is set to the bulk states of WTe₂. However, the identification of the position of the Fermi level is very unclear from the data as the feature at 0.95 eV is barely above the noise level. Is this feature reproducible? Moreover, it is unclear to me why the SHE should not be present in bulk states of WTe₂ in as much as it is allowed by symmetry and is proportional to spin-orbit coupling, which is strong in this material. Finally, is it just a coincidence that the overall amplitude of the spin accumulation in the two putative regimes of conduction only differ by a factor of about 2? The second piece of evidence is the difference in spin accumulation for the two directions of bias current. However, the authors undercut this evidence themselves by pointing out that that "WTe₂ possesses a relatively small SOC along the b-axis than the a-axis."

Response 3-1: We appreciate Reviewer 3's detailed questions on the nature of spin polarization of the hinge states. Reviewer 3 also raised a concern about other possible explanations for the spatially asymmetric spin accumulation.

First of all, we apologize that we have missed the fundamental physics of the HOTI hinge states. For the spin configuration of the hinge states, it may be worth comparing it to the conventional helical 1D edge states of 2D Z_2 topological insulators. In fact, the spatial distribution of spin-polarized electrons exhibits similarity at a certain level. For 3D helical HOTI, the hinge consists of two counterpropagating spin-up and -down modes, similar to the

helical edge of the 2D topological insulators. Recent studies of N. Shumiya *et al.* (Nat. Mater. (2022)), Y.-B. Choi *et al.* (Nat. Mater. (2020)), R. Noguchi *et al.* (Nat. Mater. (2021)) have identified one of the two topologically nontrivial hinges on the top (or bottom) surface; the hinge along the a -axis hosts a helical hinge state as for our case. In more details, followed by the earlier proposal of F. Zhang *et al.* (Phys. Rev. Lett. (2013)), it has shown that the hinge states can actually be modeled as quantum spin Hall states in the uncoupled top/bottom surface of 3D stacked structure, while the interlayer hybridization between the edges of stacked 2D TI layers cancels out the helical edges states between the top and bottom surface. Supposing that the spin-polarized electrons are injected from WTe₂ to graphene, the opposite spin polarization on the opposite hinges would match with the expected helical hinge modes of the 3D TI.

Secondly, for the question about the alternative origin of $\Delta\theta_{\mathbf{k}}$, especially the spin Hall effect (SHE), please refer to the comments by Reviewer 1 (Comment 1-2) and Review 2 (Comment 2-1), where both reviewers have also raised similar concerns. Below we present our responses to such a similar question by providing new data obtained from a newly fabricated device.

The device we fabricated to address the bulk-originating spin current issue (device #4) has a 1.5 μm gap in the middle of graphene. The optical microscopy image of the device is shown in Fig. R16a. If $\Delta\theta_{\mathbf{k}}$ originates from the hinge states, the localized $\Delta\theta_{\mathbf{k}}$ should arise only at the y -position of the WTe₂ hinge (Fig. R16b). On the other hand, if the observed $\Delta\theta_{\mathbf{k}}$ originates from the bulk spin transport in WTe₂ (i.e., due to SHE), the spin-polarized electrons injected into graphene are expected to be spread out to the left as well as to the right of the graphene area in a transverse direction to the applied electric field (Fig. R16c). Therefore, the presence of a gap in graphene would collect the accumulated spin-polarized electrons at the edge of graphene on both sides of the gap (Fig. R16c).

Figure R16. An experiment proposed to isolate the hinge characteristics (device #4). **a** An optical microscopy image is shown. Two monolayer graphene flakes are separated by a $1.5 \mu\text{m}$ gap. This device scheme is almost identical to the other devices, except for the presence of a gap in graphene. The graphene layer for the electron transport measurement is located below WTe_2 . **b, c** Schematic representation for the expected $\Delta\theta_{\mathbf{k}}$ distribution when the spin-polarized electrons in graphene originate from the WTe_2 hinge states (**b**) and when they originate from the WTe_2 bulk (**c**). Dashed rectangles are the spatial windows that we performed the Kerr-rotation measurements. The black arrows are to represent electron transport.

To check the above idea, we have investigated the magneto-optic Kerr effect on device #4. The results are shown in Fig. R17. Here we performed the measurements of the spatially resolved $\Delta\theta_{\mathbf{k}}$ at different V_G with (Fig. R17a) and without (Fig. R17b) an external magnetic field ($B_z = 1$ T). We first note that no accumulation of the spin-polarized electrons was seen on either side of the graphene gap, regardless of V_G . Secondly, with varying V_G (see Response 2-3 for the relationship between V_G and ΔE_F), Fig. R17a shows that $\Delta\theta_{\mathbf{k}}$ appears only in line with the WTe_2 hinges. We also observed a clear sign flip of $\Delta\theta_{\mathbf{k}}$ when E_F is crossing the Dirac point of the hinge states. Third, the $\Delta\theta_{\mathbf{k}}$ under magnetic field (Fig. R17b) shows the mid-gap opening of the hinge states. Under the external magnetic field B_z of 1 T, the localized $\Delta\theta_{\mathbf{k}}$ at the y -position of the hinges disappears when the Fermi level is close to the Dirac point (i.e., V_G is near 0.88 V in the case of device #4), demonstrating the lifted degeneracy of hinge eigenstates due to the broken time-reversal symmetry. To summarize, the V_G - and B_z -dependent $\Delta\theta_{\mathbf{k}}$ distribution in device #4 is essentially identical to the devices without the graphene gap (see Fig. 2 and Figs. S4, 11 in the revised manuscript). We believe these data provide evidence that SHE is not likely the origin of our observation.

Figure R17. The spatially resolved V_G -dependent $\Delta\theta_k$ measured in device #4 when $B_z = 0$ (a) and $B_z = 1$ T (b). Dashed lines indicate the edge of each graphene layer, and the black rectangle represents the multilayer WTe_2 . The direction of the applied electric field is longitudinal along the $-x$ -direction, which is parallel to the WTe_2 a -axis.

We experimentally show that SHE is not the major origin of $\Delta\theta_k$ distribution. Generally speaking, the anomalous Hall effect arises from the bulk or the surface due to the Berry curvature or scattering under spin-orbit coupled magnetization [Rev. Mod. Phys. **82**, 1539

(2010)]. It generates the transport of electrons transverse to the applied electric field, and more specifically, the transverse spin current is induced in the case of SHE. If SHE in the multilayer WTe₂ is the alternative origin of the observed $\Delta\theta_{\mathbf{k}}$, then $\Delta\theta_{\mathbf{k}}$ distribution should appear as Fig. R16c. However, the data shown in Fig. R17 indicate that this is not the case.

The way how the Berry curvature dipole affects the transverse electrical transport is different from SHE. Recent studies on the highly asymmetric WTe₂ revealed that the Berry curvature dipole is related to the nonlinear Hall effect in a non-magnetic environment [Nature **565**, 337-342 (2019); Nat. Mater. **18**, 324-328 (2019); Nat. Rev. Phys. **3**, 744-752 (2021)]. The nonlinear Hall current $\vec{j}^{2\omega}$ can be written as

$$\vec{j}^{2\omega} = \frac{e^3\tau}{2(1+i\omega\tau)} \hat{z} \times \vec{E} (\vec{\Lambda} \cdot \vec{E}),$$

where ω is the frequency, τ is the Boltzmann transport relaxation time, and Λ is the Berry curvature dipole [Phys. Rev. Lett. **115**, 216806 (2015)]. Here, the nonlinear Hall effect induced by the Berry curvature dipole is nonlinear with respect to the applied sinusoidal field, and an electrical Hall effect causes a transverse charge current rather than the transverse spin current. Hence, the $\Delta\theta_{\mathbf{k}}$ distributions observed in our experiments are not related to the nonlinear Hall effect by the Berry curvature dipole, even though the anisotropic $\Delta\theta_{\mathbf{k}}$ distribution matches that of the nonlinear Hall current. Moreover, even if the spin degree of freedom is assigned to the nonlinear Hall current for some reason, the nonlinear Hall effect should arise from the whole WTe₂ area, not exclusively from the hinge.

We have added the above discussions on the possible origins of $\Delta\theta_{\mathbf{k}}$ other than the hinge states into the revised manuscript. The discussions on the results from a newly fabricated device (device #4) are provided in the revised main text (see Fig. 4 and the related discussions), and further details about the experiments using device #4 are added in Supplementary Note 4 in the revised manuscript. The previous discussions on the orientation- and temperature-dependent $\Delta\theta_{\mathbf{k}}$ (Fig. 4 in the original version of the manuscript) are moved to the Supplementary Information (see Supplementary Notes 3-2 and 3-3).

Lastly, for the amplitude ratio of $\Delta\theta_{\mathbf{k}}$ observed in two conduction channels (i.e., along the a - and b -axis), it is just coincidental and irrelevant to the anisotropic SOC of WTe₂. The data represent that WTe₂ has the anisotropic distribution of hinge states [Nat. Mater. **19** 974-979 (2020); Nano Lett. **20**, 4228-4233 (2020)]. In preparation for the revised manuscript, we looked back at our original manuscript. There, we briefly discussed about SOC to rule out that SHE is not the origin of spin-polarized electrons in graphene; we demonstrated that there are no helical states in the hinges along the b -axis. Our idea was that if SHE is the origin of the observed $\Delta\theta_{\mathbf{k}}$,

there should be a small amount of transverse spin current and localized $\Delta\theta_k$ signal even though the SOC of WTe₂ is weak and anisotropic along the b -axis [Nat. Commun. **10**, 2044 (2019); arXiv. 2008.08785 (2020); Phys. Rev. Lett. **115**, 166601 (2015)]. In this respect, we should admit that this comparison between two conduction channels is not enough to prove the irrelevance of SHE contribution. Instead, an additional experiment using device #4 presented in the revised manuscript provides more solid evidence for the irrelevance of SHE. In the revised version, we have replaced our original discussion on the a - and b -axis channels with the experiment performed in device #4; please refer to Fig. 4 in the revised manuscript. The $\Delta\theta_k$ in device #1 with the bias voltage along the b -axis is moved to Supplementary Note 3-2 in the revised manuscript.

Comments 3-2: • Are there any flaws in the data analysis, interpretation and conclusions?
Do these prohibit publication or require revision?

In answer to this question, let me repeat my basic lack of understanding of what is being claimed here. I don't see how one explains spin accumulation at the edges of a sample without transverse spin current. Perhaps I am missing something – if this could be explained the paper would be on firmer ground. However, in view of lack of spatial resolution, the claim to direct observation should be softened.

Response 3-2: We feel sorry and apologize for not conveying the main point clearly. We believe there are the following two-fold reasons. One is about the physical fundamentals of the spinful hinge characteristics, and another is about the device structure, i.e., WTe₂/graphene heterolayer. Since the former was addressed in Response 3-1, we present a more detailed explanation of the principle of our experiment and the associated device structure in the following.

The multilayer WTe₂ is located on top of the graphene layer, where WTe₂ may be regarded as the spin-polarized electron source, and graphene can be viewed as the spin detection plane. In this way, when the spin-polarized electrons are injected from WTe₂ into graphene in the presence of the applied longitudinal electric field, we optically detect such electrons by measuring the Kerr rotation angle of the incident laser beam. Of course, the spatially distributed Kerr rotation signal does not directly represent the WTe₂ hinge characteristics. Rather the information of spin-polarized electrons (i.e., the hinge characteristics) is imprinted within a finite spatial dimension near the WTe₂. Here, the “spatial dimension” is limited mainly by the spin diffusion length of graphene. Therefore, $\Delta\theta_K$ observed in our experiments is not an intrinsic signal from the geometric edge; it shows the spin-polarized electrons that are localized within a finite spin diffusion length.

To be more specific, there are the following technical reasons why we used graphene to detect the signature of the hinge states. First, the spatial length scale of the hinge states, whose width is only about a few nanometers [Nano Lett. **20**, 4228-4233 (2020); Nat. Phys. **14**, 918-924 (2018)], is far smaller than the spot size of the incident laser beam. This means that any optical signals originating from the hinge states would be extremely negligible compared to that from the bulk, surface, or other background signals generated within the beam spot. This scaling problem limits the signal-to-noise ratio and hinders the detection of the hinge state if $\Delta\theta_K$ were to be measured directly on the region where the laser is excited. Secondly, because the hinge, by definition, is a junction between two surfaces, an incident laser introduces both the in-plane and the out-of-plane excitation at the two surfaces facing the hinge. Additionally, the hinge

induces an anisotropic polarization-dependent reflection in the photoexcited area, thereby limiting the clear isolation of $\Delta\theta_k$ from the possible spin-polarized state.

To minimize the geometric effects of the hinge described above, we take advantage of graphene. Graphene is known to exhibit a long spin diffusion length of up to 30 μm due to the weak spin-orbit coupling and high mobility [Nano Lett. **16**, 3533-3539 (2016); Nat. Commun. **6**, 6766 (2015); Nat. Nanotech. **9**, 794-807 (2014)]. Under the electric field applied to the device, the electric potential gradient generates a longitudinal electron transport that injects electrons from WTe_2 to graphene, as described in Supplementary Note 2. The spin polarization of the injected electrons can be maintained long enough to be detected by the optical measurements due to the long spin diffusion length of graphene. Therefore, because $\Delta\theta_k$ is distributed into the graphene rather than the localized region near the WTe_2 hinges, we have chosen a line cut at $x = 0.75 \mu\text{m}$ to analyze $\Delta\theta_k$ while keeping a certain distance from the edge of the WTe_2 crystal.

Based on the above experimental design, the main claim of our investigation can be stated as follows. The spin-polarized electrons were concentrated at the hinges of multilayer WTe_2 because there are 1D boundaries (hinges) between the gapped surfaces in 3D HOTI. Recent studies on a 3D HOTI Bi_4Br_4 claim the spinful hinge state can be understood as an uncoupled quantum spin Hall state, which we have discussed the fundamentals of the hinge states in Response 3-1 [N. Shumiya *et al.*, Nat. Mater. (2022)]. Unfortunately, theoretical work [Phys. Rev. Lett. **123**, 186401 (2019)] on multilayer WTe_2 has only dealt with a spinless model, and no experimental studies have been reported on the spinful hinge states in 3D HOTI WTe_2 . What is certain, though, is that the multilayer WTe_2 contains both in-plane mirror and inversion asymmetry that support the out-of-plane spin orientation. Following this idea, we have focused on the possibility of the existence of topologically protected spinful hinges. Indeed, we believe that our MOKE spectroscopy provides experimental evidence of the existence of a spin-polarized state that is protected by time-reversal symmetry. As discussed in Response 3-1, additional experiments using device #4 (a newly fabricated device with a graphene “gap”) have evidenced no bulk effect (e.g., spin Hall effect), implying the origin of observed spin polarization is not a transverse spin transport arising from the bulk WTe_2 .

The last point advised by Reviewer 3 is to what extent we can claim “direct observation”. Because our MOKE does not directly probe the hinge states with an atomic spatial resolution, we agree that such a title would be misleading to the readers. Following Reviewer 3’s advice, we have modified the title of our manuscript, where we eliminated “direct” in the title as well as in the main text.

Comments 3-3:

- Is the methodology sound? Does the work meet the expected standards in your field?

I worries me a bit that many of the observations such as the neutrality points for the graphene and WTe₂, and the polar Kerr signal itself, are so close to the noise limit. It would be useful to see some evidence for the reproducibility of these effects. Another point regarding methodology is that I don't understand why the polar Kerr effect is observed in the graphene rather than in the WTe₂ itself.

- Is there enough detail provided in the methods for the work to be reproduced?

I believe there is enough detail in this regard.

Response 3-3: We thank Reviewer #3 for his/her careful examination on the electronic transport as well as on the origin of the Kerr effect. Reviewer #3 also provides his/her advice on the reproducibility of the data.

For the electronic transport, we admit that Fig. 2 and Fig. S1 in the original manuscript are somewhat confusing. Knowledge of the charge neutral point is very important to confirm the band alignment of WTe₂ and graphene. To account for this point, we have fabricated three more devices and checked the V_G -dependent drain current, as well as performed a self-consistency check. Please note that both Reviewer #1 and #2 raised similar concerns; we may reuse our responses here to accommodate the concern.

The optical microscope images of all devices used in our study are shown in Fig. R18. Figure R19 shows the thickness and the number of layers checked by the AFM measurements. The electronic transport data, including the original device, are summarized in Fig. R20. Please note that for all four devices, not only the electronic transport but also MOKE spectroscopy was performed, and the results are well-reproduced (Fig. R21). The electronic transport data from all devices are organized in Fig. S3 in the revised manuscript. For the set of MOKE data, please refer to Figs. 1, 2, and Figs. S1, S5-11 for device #1, Fig. S12 for devices #2 and #3, and Fig. S12 for device #4 in the revised manuscript.

Device	WTe ₂ thickness (nm)	Number of layers	V _{CNP} (V)
device #1	4	5	0.95
device #2	16	~20	1
device #3	3	4	0.92
device #4	5	7	0.88

Table R3. Thickness and V_{CNP} of each device used in the experiments. The thickness was measured by AFM (Fig. R19), and V_{CNP} was obtained from the V_{G} -dependent electrical transport (Fig. R20).

Figure R18. Optical microscopy images of devices #1-4. **a.** Device #1 is used to obtain the data in Figs. 1-3 in the revised manuscript. **b, c.** Devices #2 and #3 are newly fabricated devices with the same device structure as device #1. **d.** Device #4 has a 1.5 μm wide gap in the middle of graphene. All electrodes in the devices were made using 5 nm Ti and 25 nm Au. Numbers 1-4 in the figures are the contact indices. The black arrows in each figure show the orientation of WTe₂ crystal axes.

Figure R19. a-d. Thickness of the multilayer WTe_2 flake in the devices measured by AFM. The thickness of the metal contact (Ti 5 nm, Au 25 nm) close to WTe_2 is also measured for reference.

Figure R20. a-d V_G -dependent electrical characteristics obtained from devices #1-4. Contact 1-3 and 2-4 stand for the source and drain contact for each device. The current between contact 1, 3 (2, 4) corresponds to the current along the a - (b -) axis of each device.

For the methodology of why the signals were measured in graphene (rather than directly on the hinges), we have provided somewhat details on this concern in Response 3-2. Here we will briefly summarize the contents. First, there is a spatial difference in scale between the hinges and the spot size of the incident beam. Because the 1D hinge is a junction between two 2D surfaces, the finite size of laser excitation (on the order of μm) is much larger than the hinge (on the order of nm). Second, the geometric nature of the 1D hinge in 3D crystal prevents the precise measurements of the rotated light polarization. Additionally, please note that the hinge state of WTe_2 exists regardless of the applied longitudinal electric field because the bulk-boundary correspondence is protected by the inversion and the time-reversal symmetry. Therefore, even if there was an observable signature under laser excitation directly on the hinges, the Kerr rotation signals will be canceled out after subtracting the spatially resolved Kerr rotation distributions with and without the bias voltage, i.e., resulting in a net zero $\Delta\theta_K$.

Figure R21. Spatially resolved V_G -dependent $\Delta\theta_K$ from device #2 (a) and device #3 (b).

In the revised manuscript, we have included a more detailed discussion. Table R3 and Figs. R18-21 are added to the Supplementary Information as Table S1 and Figs. S2-4 and Fig. S13, respectively. Corresponding descriptions of the data from devices with various thickness is added in Supplementary Note 3-4.

Reviewers' Comments:

Reviewer #1:

Remarks to the Author:

The authors' reply is satisfactory. The revised manuscript added new content, and has made significant improvements over the previous version. Therefore, I would like to suggest it be published in Nature Communications.

Reviewer #2:

Remarks to the Author:

In my first review of this paper, I had commented that the manuscript was "very intriguing, interesting, and puzzling" and that it was an "elegant experiment, ambitious in its goal, and it would be worthwhile publishing this data in a high profile journal such as Nature Communications." However, I raised several deficiencies in that version which needed to be addressed.

1. The authors needed to show evidence that would rule out the conventional spin Hall effect or at least address why this was not a likely explanation. The authors have addressed this concern in two ways. First, by pointing out that the relationship between the applied electric field and the spatial distribution of the spin-polarized electrons (as detected through the Kerr signal) was inconsistent with the spin Hall effect. They also carried out measurements on a fourth device with a different geometry, one with a gap in the middle of the graphene. This device would allow one to distinguish between spin-polarized electrons originating from the bulk spin Hall effect and from spinful hinge states. The observations appear to be consistent with the latter. I find these arguments quite convincing.
2. I was very concerned about the paper drawing conclusions solely from measurements on a single device. The authors have now shown data from 2 additional devices in the same configuration and this reproduces the original data. In addition, they fabricated a control device (#4) that provides additional convincing data.
3. I had also mentioned some issues with the confusing aspects of one of the figures (Fig. 4) and various issues with the language and style in the paper. The authors have addressed both these concerns.

In summary, the revised paper is a significant improvement over the initial submission and makes a strong case for the principal claim made in the paper, namely experimental evidence for spinful hinge states in WTe₂ using spatially resolved Kerr spectroscopy. I think that this rises to the standards of Nature Communications.

Point-by-point responses to the issues raised by the reviewers

General remarks and comments of Reviewer 1:

The authors' reply is satisfactory. The revised manuscript added new content, and has made significant improvements over the previous version. Therefore, I would like to suggest it be published in Nature Communications.

Response 1 general remarks: We appreciate that Reviewer 1 found our revised manuscript satisfactory and suggested publication. Thanks to the comments provided by the Reviewer 1, we were able to improve the revised manuscript both qualitatively and quantitatively. While preparing the final version of the manuscript, we revised and corrected a few grammatical expressions to improve the readability of the manuscript.

Below we listed the edited sentences.

Edited contents

- Corresponding author email address is added

- Methods, Data Availability, References, Acknowledgements, Author Contributions, and Competing Interests sections are arranged according to the formatting instructions.

- Abstract
 T_d -WTe₂ is a promising candidate to reveal topological hinge excitation in an atomically thin regime.

- Main text
(page 3, line 13)
Among such candidates, WTe₂ has recently attracted much interest in investigating the electronic correlations as well as exploring the topologically protected quantum phenomena.

(page 4, line 16)
Multilayer T_d -WTe₂ has a non-centrosymmetric orthorhombic structure belonging to the SG 31 ($Pmn2_1$) space group with two perpendicular axes (a - and b -axis) and one mirror line along the b -axis (Fig. 1a)

(page 5, line 3)

~ with a sub-micrometer-scale resolution.

(page 5, line 21)

~ while $\Delta\theta_{\mathbf{k}}$ is evenly distributed throughout $|y| \leq 1.85 \mu\text{m}$ at $V_G = -1$ and 2 V .

(page 6, line 1)

~ the observed $\Delta\theta_{\mathbf{k}}$ distributions at $V_G = 0$ and 1 V match the spin-polarized in-gap states localized in the hinge, ~

(page 6, line 4)

To elucidate the bulk- and hinge-originated $\Delta\theta_{\mathbf{k}}$ in detail, ~

(page 6, line 16)

. Note that the multilayer WTe_2 is ~

(page 6, line 18)

As for the HOTI characteristics, we note that the band topology of the multilayer WTe_2 should be protected by the time-reversal symmetry.

(page 6, line 23)

In Fig. 3a, where B_z is 0.5 T , we see that the $\Delta\theta_{\mathbf{k}}$ near the hinges vanishes as V_G approaches the charge neutrality. With increasing B_z of 1 T (Fig. 3b), the localized $\Delta\theta_{\mathbf{k}}$ survives only when V_G is pushed further below (0 V) and above (1.5 V) the charge neutrality point.

(page 7, line 7)

Without B_z , the hinge states remain gapless because the degeneracy of the Dirac point is protected by time-reversal symmetry.

(page 8, line 2)

In our experiment regime, the applied B_z makes the hinge a boundary between one parallel to B_z and another perpendicular to B_z .

(page 8, line 6)

~ i.e., spin Hall effect (SHE), ~

(page 8, line 11)

~ a 1.5 μm wide gap ~

(page 8, line 19)

Here we measured the spatially resolved $\Delta\theta_{\mathbf{k}}$ at different V_G with and without an external magnetic field ($B_z = 1$ T) (see Supplementary Note 4 and Figs. S15, 16 for spatially resolved $\Delta\theta_{\mathbf{k}}$ with B_z).

- Figure captions

(Figure 1b)

The pump laser with a spot size of 1.5 μm ~

(Figure 1d)

A normalized polar plot of the polarization-dependent absorption of the multilayer WTe_2 is shown.

(Figure 4)

Spatially resolved differential Kerr rotation on a device with a spatial gap in graphene.

(Figure b,c)

Schematic diagrams of the expected $\Delta\theta_{\mathbf{k}}$ when the spin-polarized electrons are injected ~

General remarks and comments of Reviewer 2:

In my first review of this paper, I had commented that the manuscript was “very intriguing, interesting and puzzling” and that it was an “elegant experiment, ambitious in its goal, and it would be worthwhile publishing this data in a high profile journal such as Nature Communications.” However, I raised several deficiencies in that version which needed to be addressed.

1. The authors needed to show evidence that would rule out the conventional spin Hall effect or at least address why this was not a likely explanation. The authors have addressed this concern in two ways. First, by pointing out that the relationship between the applied electric field and the spatial distribution of the spin-polarized electrons (as detected through the Kerr signal) was inconsistent with the spin Hall effect. They also carried out measurements on a fourth device with a different geometry, one with a gap in the middle of the graphene. These devices would allow one to distinguish between spin-polarized electrons originating from the bulk spin Hall effect and from spinful hinge states. The observations appear to be consistent with the latter. I find these arguments quite convincing.
2. I was very concerned about the paper drawing conclusions solely from measurements on a single device. The authors have now shown data from 2 additional devices in the same configuration and this reproduces the original data. In addition, they fabricated a control device (#4) that provides additional convincing data.
3. I had also mentioned some issues with the confusing aspects of one of the figures (Fig. 4) and various issues with the language and style in the paper. The authors have addressed both these concerns.

In summary, the revised paper is a significant improvement over the initial submission and makes a strong case for the principal claim made in the paper, namely experimental evidence for spinful hinge states in WTe₂ using spatially resolved Kerr spectroscopy. I think that this rises to the standards of Nature Communications.

Response: We are grateful that Reviewer 2 is well aware of the main idea we made in the revised manuscript. We are also happy that that Reviewer 2 considered our revised manuscript to be matched the standards of Nature Communications. Please note that while preparing the final version of the manuscript, we revised and corrected a few grammatical expressions to improve the readability of the manuscript.

Below we listed those edited sentences.

Edited contents

- Corresponding author email address is added
- Methods, Data Availability, References, Acknowledgements, Author Contributions, and Competing Interests sections are arranged according to the formatting instructions.

- Abstract

T_d -WTe₂ is a promising candidate to reveal topological hinge excitation in an atomically thin regime.

- Main text

(page 3, line 13)

Among such candidates, WTe₂ has recently attracted much interest in investigating the electronic correlations as well as exploring the topologically protected quantum phenomena.

(page 4, line 16)

Multilayer T_d -WTe₂ has a non-centrosymmetric orthorhombic structure belonging to the SG 31 ($Pmn2_1$) space group with two perpendicular axes (a - and b -axis) and one mirror line along the b -axis (Fig. 1a)

(page 5, line 3)

~ with a sub-micrometer-scale resolution.

(page 5, line 21)

~ while $\Delta\theta_{\mathbf{k}}$ is evenly distributed throughout $|y| \leq 1.85 \mu\text{m}$ at $V_G = -1$ and 2 V.

(page 6, line 1)

~ the observed $\Delta\theta_{\mathbf{k}}$ distributions at $V_G = 0$ and 1 V match the spin-polarized in-gap states localized in the hinge, ~

(page 6, line 4)

To elucidate the bulk- and hinge-originated $\Delta\theta_{\mathbf{k}}$ in detail, ~

(page 6, line 16)

. Note that the multilayer WTe₂ is ~

(page 6, line 18)

As for the HOTI characteristics, we note that the band topology of the multilayer WTe₂ should be protected by the time-reversal symmetry.

(page 6, line 23)

In Fig. 3a, where B_z is 0.5 T, we see that the $\Delta\theta_{\mathbf{k}}$ near the hinges vanishes as V_G approaches the charge neutrality. With increasing B_z of 1 T (Fig. 3b), the localized $\Delta\theta_{\mathbf{k}}$ survives only when V_G is pushed further below (0 V) and above (1.5 V) the charge neutrality point.

(page 7, line 7)

Without B_z , the hinge states remain gapless because the degeneracy of the Dirac point is protected by time-reversal symmetry.

(page 8, line 2)

In our experiment regime, the applied B_z makes the hinge a boundary between one parallel to B_z and another perpendicular to B_z .

(page 8, line 6)

~ i.e., spin Hall effect (SHE), ~

(page 8, line 11)

~ a 1.5 μm wide gap ~

(page 8, line 19)

Here we measured the spatially resolved $\Delta\theta_{\mathbf{k}}$ at different V_G with and without an external magnetic field ($B_z = 1$ T) (see Supplementary Note 4 and Figs. S15, 16 for spatially resolved $\Delta\theta_{\mathbf{k}}$ with B_z).

- Figure captions

(Figure 1b)

The pump laser with a spot size of 1.5 μm ~

(Figure 1d)

A normalized polar plot of the polarization-dependent absorption of the multilayer WTe₂ is shown.

(Figure 4)

Spatially resolved differential Kerr rotation on a device with a spatial gap in graphene.

(Figure b,c)

Schematic diagrams of the expected $\Delta\theta_K$ when the spin-polarized electrons are injected ~